# The Challenge of Reliable Vision–Language Model Responses in Driving

## Abstract

A reliable driving assistant should provide consistent responses and reasoning based on observed information. In this work, we investigate whether Vision-Language Models (VLMs), when applied as driving assistants, can response consistantly and genuinely understand how present observations shape future outcomes, or whether their outputs merely reflect patterns memorized during training without grounded temporal reasoning. While recent efforts have integrated VLMs into autonomous driving, prior studies typically emphasize scene understanding and instruction generation, implicitly assuming that strong visual interpretation naturally enables consistant future reasoning and thus ensures reliable decision-making, a claim we critically examine. We focus on two major challenges limiting VLM reliability in this setting: response inconsistency, where minor input perturbations yield different answers or, in some cases, responses degenerate toward near-random guessing, and limited temporal reasoning, in which models fail to reason and align sequential events from current observations, often resulting in incorrect or even contradictory responses. Moreover, we find that models with strong visual understanding do not necessarily perform best on tasks requiring temporal reasoning, indicating a tendency to over-rely on pretrained patterns rather than modeling temporal dynamics. To address these issues, we adopt existing evaluation methods and introduce FutureVQA, a human-annotated benchmark dataset specifically designed to assess future scene reasoning. In addition, we propose a simple yet effective self-supervised tuning approach with chain-of-thought reasoning that improves both consistency and temporal reasoning without requiring temporal labels.
*The data and code for our experiments will be released upon acceptance.*

## 1 Introduction

Modern Vision-Language Models (VLMs) exhibit human-like perception and reasoning capabilities, enabling more natural and intelligent interactions in everyday applications (Liu et al., 2023b;a; Wang et al., 2023; Liu et al., 2024; Bai et al., 2023; Young et al., 2024; Zhang et al., 2023b). Recent studies (Jiang et al., 2024; Hwang et al., 2024; Fu et al., 2025; Renz et al., 2025) have explored their potential as driving assistants, applying them to scene analysis and decision-making in complex driving environments. Trained on large-scale visual data, these models demonstrate strong abilities in interpreting visual cues and traffic signs, generating high-level driving instructions that resemble human reasoning and can assist autonomous vehicles.

However, despite these encouraging advancements, most existing approaches implicitly assume that strong visual understanding naturally translates into reliable future scene prediction and reasoning. In this work, we critically examine this assumption by evaluating the consistency and reliability of VLM responses in driving scenarios. Specifically, we investigate whether these responses stem from genuine temporal reasoning or merely reflect memorized knowledge acquired during pre-training (Fatemi et al., 2024; Xu et al., 2024).

Specifically, we address the following challenges: (1) Response inconsistency, where identical or nearly identical inputs can lead to divergent or unstable outputs; and (2) limited temporal reasoning, where the model fails to maintain coherent reasoning across events that unfold over time, often producing incorrect predictions or even contradictory answers to follow-up questions requiring tem-

Figure 1: Reliability failures in VLMs. The figure illustrates three issues: (i) response inconsistency—identical or very similar prompts yield different answers; (ii) contradiction—correct local interpretation but inconsistent future description; and (iii) temporal misalignment—events predicted at incoherent times despite accurate per-frame cues.

poral understanding. These issues highlight a fundamental limitation: the model's lack of temporal grounding. Unlike humans, VLMs do not experience the flow of time and may over-rely on memorized patterns from pretraining rather than performing genuine temporal reasoning (Fatemi et al., 2024).

Our experiments show that both open-source and commercial VLMs exhibit varying degrees of inconsistency when answering driving-related questions, even under minimal input perturbations—such as shuffling the order of answer options in a visual question answering (VQA) task. Furthermore, we find that models with stronger visual understanding are not necessarily better at reasoning about future scenes or events. In some cases, these models perform worse than others, revealing a disconnect between visual perception and temporal reasoning. These findings underscore a critical concern: the potential risks of deploying VLMs in safety-critical applications such as autonomous driving, where consistent and temporally grounded reasoning is essential.

Alongside standard evaluation methods, we introduce FutureVQA, a fully human-annotated benchmark designed to assess how well VLMs can reason about future scenes based on their understanding of preceding visual observations. In addition, we propose a simple yet effective self-supervised tuning approach that improves the model's ability to perform consistent temporal reasoning and scene prediction—without requiring explicit temporal labels.

In summary, our main contributions include: (1) We identify and analyze key limitations of current Vision-Language Models (VLMs) in driving scenarios, including response inconsistency and lack of temporal reasoning, which pose risks for safety-critical applications. (2) We introduce **FutureVQA**, a human-annotated benchmark designed to evaluate VLMs' ability to reason about future scenes based on prior visual context. (3) We propose a simple yet effective self-supervised tuning method that enhances temporal consistency and future scene prediction without requiring temporal supervision.

## 2 RELATED WORK

**Vision Language Models:** Recent advances in LLMs have greatly expanded the scope of multimodal research. In the visual domain, models like LLaVA (Liu et al., 2023b;a), QWen (Bai et al., 2023), Yi-VL (Young et al., 2024), and CogVLM (Wang et al., 2023) have made significant strides in image-text reasoning, offering detailed analyses of visual data alongside textual descriptions. LLaVA-Next (Liu et al., 2024) further enhances this capability by supporting higher-resolution inputs, enabling more detailed image understanding. For video-text understanding, models such as Video-LLaMA (Zhang et al., 2023b) and LLaVA-Video (Zhang et al., 2024) have advanced narrative comprehension by incorporating temporal information from dynamic visual content. Beyond generic VLMs, modern VLMs are increasingly integrated into autonomous driving (Nie et al., 2024; Chen et al., 2024b; Liao et al., 2024; Pan et al., 2024; Gopalkrishnan et al., 2024; Zhou et al., 2024a; You et al., 2024; Chen et al., 2024a; Sima et al., 2023; Wang et al., 2024a), enhancing scene understanding and decision-making.

**Future Scene Reasoning:** Predicting future scenes is a crucial task in robotics and autonomous driving, requiring models to understand the physical world and how scenes evolve over time. Recently, the construction of world models has gained popularity across various modalities, including point cloud generation, which aims to construct a realistic 3D representation of the world over time (Khurana et al., 2023; Huang et al., 2024; Yang et al., 2024b; Manivasagam et al., 2020), and video generation under different environmental conditions and control signals (Wang et al., 2024b; Zhao et al., 2024; Gao et al., 2024; Hu et al., 2023; 2024; Zhou et al., 2024b; Wang et al., 2024c; Jia et al., 2023; Hassan et al., 2024).

**Visual Question Answering:** Early VQA datasets primarily focused on general image-question performance (Antol et al., 2015; Zhang et al., 2016; Goyal et al., 2017). Beyond general-purpose VQA, researchers have explored domain-specific applications, such as medical VQA (Lau et al., 2018; Bae et al., 2024) and science-driven VQA (Kembhavi et al., 2017). To provide a more robust evaluation, some datasets go beyond free-form sentence answers and adopt structured answer formats, such as Yes/No questions (Fu et al., 2024) and multiple-choice formats (Liu et al., 2023d; Wu et al., 2024; Fu et al., 2024), ensuring a more consistent and objective assessment of model performance. Recently, VQA in autonomous driving has gained attention, aiming to enhance scene understanding in dynamic traffic environments (Sachdeva et al., 2024; Malla et al., 2023; Sima et al., 2023; Wang et al., 2024a; Qian et al., 2023; Deruyttere et al., 2019; Vasudevan et al., 2018).

## 3 PROBLEM FORMULATION AND EVALUATION

A reliable safe-driving assistant should anticipate how actions and events unfold over time, remain temporally coherent, and respond consistently under semantics-preserving prompt changes. Let $V_t = \{I_i \mid i \leq t\}$ be the historical frames up to time $t$. A VLM $\psi$ is queried to produce a description $a_{t+\Delta t}$ of the scene at time $t + \Delta t$, where $\Delta t \in \mathbb{Z}^+$ is the prediction horizon. Using the model's response when given the ground-truth future frame $I_{t+\Delta t}$ as a reference, reliability requires alignment between past-only and future-conditioned predictions:

$$P_\psi(a_{t+\Delta t} \mid V_t) \approx P_\psi(a_{t+\Delta t} \mid I_{t+\Delta t}). \tag{1}$$

### 3.1 RESPONSE UNRELIABILITY AND INCONSISTENCY

Existing studies on language models highlight reliability issues, including hallucination (Kalai et al., 2025) and sensitivity to input phrasing (Ahn & Yin, 2025). These concerns are acute in autonomous driving, where decisions must rest on consistent and trustworthy reasoning. For the multiple-choice VQA variant with input $x$ and $K$ options, the VLM $\psi$ induces a categorical distribution $P_\psi(k \mid x)$ over answers $k \in \{1, \dots, K\}$. We consider semantics-preserving perturbations $T_\sigma(x)$ such as shuffling options by permutation $\sigma \in S_K$, and align labels via $\tilde{P}_\psi(k \mid T_\sigma(x)) := P_\psi(\sigma(k) \mid T_\sigma(x))$.

One potential source of inconsistency is *prompt-perturbation sensitivity*, which we describe as a distributional shift under such perturbations, measured by a nonzero total-variation (TV) distance:

$$\text{TV}\big(P_\psi(\cdot \mid x), \tilde{P}_\psi(\cdot \mid T_\sigma(x))\big) = \tfrac{1}{2} \sum_{k=1}^{K} \big|P_\psi(k \mid x) - \tilde{P}_\psi(k \mid T_\sigma(x))\big| > 0. \tag{2}$$

Another manifestation is a change in the top-1 prediction under perturbations, denoted as the flip rate (FR) with ties broken by the smallest index:

$$\text{FR}(x) := \Pr_{\sigma \sim \text{Unif}(S_K)} \Big[ \arg\max_k P_\psi(k \mid x) \neq \arg\max_k \tilde{P}_\psi(k \mid T_\sigma(x)) \Big], \tag{3}$$

Another potential source of inconsistency is *random guessing*: even when Equation (2) and Equation (3) are (near) zero, repeated runs may differ because the model samples from a near-uniform distribution. In this regime, predictions have accuracy $\approx 1/K$, entropy $\approx \log K$, self-agreement $R_2(x) = \sum_{k=1}^{K} P_\psi(k \mid x)^2 \approx 1/K$, and are invariant to semantics-preserving perturbations (i.e., $\text{TV} \approx 0$ and $\text{FR}(x) = 0$ at the distribution level with a fixed tie-break). In Section 5, we show that—regardless of model size—both open-source and commercial VLMs exhibit accuracy drops and elevated flip rates under option shuffles with the question fixed, consistent with distributional shifts rather than uniform guessing.

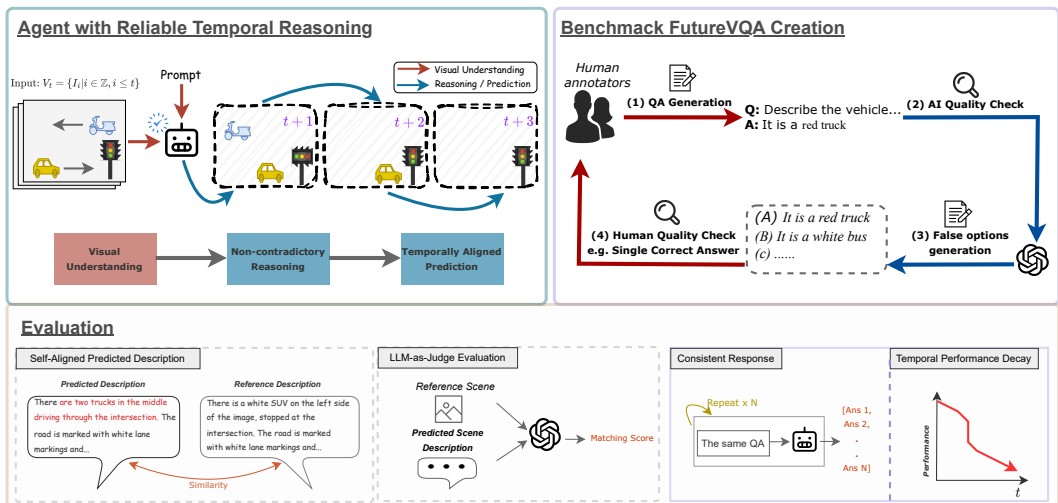

Figure 2: Overview of our framework for evaluating reliable temporal reasoning in VLM driving assistants. **Left:** The agent consumes past frames $V_t$ and a prompt to generate temporally aligned predictions over a *variable* future horizon. **Right (FutureVQA):** Benchmark construction combines human and AI contributions: human experts create natural Q/A pairs, while AI performs quality control to ensure answerability and consistency. **Bottom (Evaluation):** To thoroughly analyze model reliability, we adopt a self-aligned future description setup, where a model's predicted description is compared to a reference response generated by the same model when the actual future frames are provided. An AI checker is further applied to validate that predictions remain coherent and meaningful. Beyond this, we evaluate consistency under repeated queries and option shuffling, and analyze temporal performance decay to quantify how model reliability changes as the prediction horizon increases.

## 3.2 CONTRADICTION AND TEMPORAL MISALIGNMENT

Despite VLMs' ability to accurately interpret current traffic conditions, they often produce contradictory descriptions when reasoning about future scenes. As shown in Figure 1, a model may correctly identify visual cues and vehicle intentions at the current time based on the input, yet fail to answer follow-up questions consistently. These contradictions suggest that rich and accurate visual interpretation alone does not equip VLMs with the ability to reason about how a scene may evolve over time. In other words, while the model may learn associations between images and their corresponding textual descriptions, it does not genuinely understand their real-world implications or how present actions influence future outcomes. Another issue is temporal misalignment. As shown in Figure 1, VLMs may correctly interpret visual cues and identify individual events, yet they often fail to align these within a coherent temporal structure, as they do not experience time flow as humans do. This limitation is especially critical in driving scenarios, where outcomes such as collisions depend not only on the intentions of surrounding agents but also on the precise timing of their movements.

**Formalization.** Let $\mathcal{A}$ be the response space and $\mathcal{R}_{t+\Delta} \subseteq \mathcal{A}$ the set of admissible (reference) responses for time $t+\Delta$. Consider two tasks that condition on different information sets:

$$\psi_{\text{pred}}^{\star} := \arg\max_{\psi} \ \mathbb{E}\big[P_{\psi}(\mathcal{R}_{t+\Delta} \mid V_t)\big], \qquad \psi_{\text{ref}}^{\star} := \arg\max_{\psi} \ \mathbb{E}\big[P_{\psi}(\mathcal{R}_{t+\Delta} \mid V_{t+\Delta})\big], \qquad (4)$$

where $V_t$ is the history up to $t$ and $V_{t+\Delta}$ denotes the future slice at $t+\Delta$. In general, these Bayes-optimal solutions are *not necessarily the same*—they are not guaranteed to coincide:

$$\psi_{\text{pred}}^{\star} \ \neq \ \psi_{\text{ref}}^{\star} \qquad (5)$$

Empirically (Section 5), we observe behavior consistent with this non-equivalence: models that perform well when directly shown $V_{t+\Delta}$ can still contradict themselves across follow-ups and exhibit temporal misalignment when forecasting from $V_t$ alone.

Figure 3: Example of the FutureVQA task. The VLM is asked to answer questions about future scenes based on predictions, without access to the corresponding future frames.

---

**Algorithm 1** Self-Aligned Future Description

**Require:** Model $\psi$, Visual Input $V_t = \{I_i \mid i \leq t\}$, horizon $\Delta t \in \mathbb{Z}^+$, similarity/quality measure $\mathcal{M}(\cdot, \cdot)$, threshold $\tau$
1: $a^{\text{pred}}_{t+\Delta t} \leftarrow \psi(V_t, \Delta t)$ {Predicted *response* at $t+\Delta t$ from history}
2: $a^{\text{ref}}_{t+\Delta t} \leftarrow \psi(\{I_{t+\Delta t}\}, 0)$ {Reference *response* using actual future frame}
3: $q \leftarrow \mathcal{M}\big(a^{\text{pred}}_{t+\Delta t}, a^{\text{ref}}_{t+\Delta t}\big)$
4: **return** $q$

---

**Algorithm 2** Multi-trial Evaluation for Consistency

**Require:** Model $\psi$, Question $Q$, Visual Input $V_t$, Answer $A$, Number of Trials $N$
1: **for** $i = 1$ to $N$ **do**
2: $\quad Q_i \leftarrow$ SHUFFLEOPTIONS($Q$)
3: $\quad P_i \leftarrow \psi(V_t, Q_i)$
4: $\quad$ **if** $P_i \neq A$ **then**
5: $\quad\quad$ **return** False
6: $\quad$ **end if**
7: **end for**
8: **return** True

---

## 3.3 EVALUATION AND METRICS

This section turns the reliability criteria from Section 3 into practical tests. Since comparing full distributions $P_\psi(\cdot \mid \cdot)$ is impractical, we use paired queries and controlled perturbations as proxies. We evaluate (i) self-alignment between past-only predictions and future-conditioned references, (ii) stability to semantics-preserving prompt changes (paraphrases, option shuffles with label alignment), and (iii) behavior across horizons $\Delta t$.

**Self-Aligned Future Description.** As in Algorithm 1, we test whether a model's description of the future scene based on past context $V_t$ aligns with the description it produces when directly given the future slice $V_{t+\Delta t}$. We compare the predicted response $a^{\text{pred}}$ with the reference response $a^{\text{ref}}$ using a similarity measure $\mathcal{M}$. A conventional choice for $\mathcal{M}$ is to adopt statistical metrics developed for machine translation (Papineni et al., 2002; Lin, 2004; Banerjee & Lavie, 2005; Vedantam et al., 2014; Anderson et al., 2016). Typical examples include BLEU (Papineni et al., 2002) and ROUGE (Lin, 2004), which compute n-gram overlaps between sentences. This general family can be expressed as

$$\mathcal{M}_{\text{n-gr}}(a^{\text{pred}}, a^{\text{ref}}) = f\left(\frac{\sum_{n=1}^{N} w_n \cdot g_n(a^{\text{pred}}, a^{\text{ref}})}{\sum_{n=1}^{N} w_n}\right) \cdot BP, \qquad (6)$$

where $g_n$ denotes an n-gram similarity function weighted by $w_n$, $f(\cdot)$ applies a transformation (e.g., geometric mean), and $BP$ is a brevity penalty to adjust for length differences.

**LLM-as-Judge Evaluation.** While widely used for evaluating language models, statistical methods struggle to capture in-depth spatial relationships (Chang et al., 2024a) and the complex semantic meanings (Zheng et al., 2023) handled by modern models. An alternative is *model-based evaluation* (Liu et al., 2023c; Zheng et al., 2023; Fu et al., 2023b; Yuan et al., 2021; Sellam et al., 2020; Chang et al., 2024a;b), which leverages an advanced judge model $\mathcal{J}_m$ to assess response quality. In this setup, $\mathcal{J}_m$ is prompted to rate a response based on the visual input $x_t$, producing a score in the range $\mathcal{J}_m(a^{\text{pred}}, x_t) \in \mathbb{Z} \cap [1, 10]$. In our experiments we use GPT-4o as the judge, with details and prompt templates provided in Appendix D.

**FutureVQA Benchmark.** To complement existing evaluation metrics and address their limitations in capturing temporal reasoning and visual dynamics, we introduce the **FutureVQA Benchmark** (Figure 2)—a dataset comprising 2.7k manually annotated question-answer pairs. While existing datasets such as DriveLM (Sima et al., 2023) contribute to general scene understanding, they do not explicitly challenge VLMs on time-specific future prediction. Moreover, many rely on structured templates or rule-based generation, which limits the diversity and naturalness of question formats. In contrast, our dataset is constructed by human expert annotators based on individual video clips, featuring diverse and naturally phrased questions tailored to each scene. See Figure 3 and Algorithm 2 for the benchmark examples and the multi-trial protocol. For a detailed comparison and an overview of the dataset's contributions, please refer to Appendix A.

We evaluate performance across horizons from $t+1$ to $t+12$ seconds using accuracy (%). To capture both pointwise and temporal trends, we report: (i) **Acc@t**, accuracy at horizon $t$, reflecting prediction capability at different time steps; (ii) **$\Delta$Acc$_{1s}^{12s}$**, the accuracy drop between $t+1$ and $t+12$, indicating performance decay; (iii) **mAcc$_{(1 \rightarrow 12s)}$**, mean accuracy over horizons 1–12, summarizing overall performance; (iv) **Normalized Drop Ratio (NDR)**, defined as NDR $= \frac{1}{\eta_0} \sum_{t=1}^{T} (\eta_{t-1} - \eta_t)$, the cumulative accuracy drop normalized by the initial value $\eta_0$, where $\eta_t$ denotes accuracy at horizon $t$; and (v) **Mean Relative Accuracy Retention (mRAR)**, mRAR $= \frac{1}{T} \sum_{t=1}^{T} \frac{\eta_t}{\eta_0}$, the average ratio of accuracy at each horizon relative to the initial value.

# 4 FUTUREAGENT: AN APPROACH FOR ENHANCED TEMPORAL REASONING

To address the limitations in temporally grounded reasoning, we propose a self-supervised fine-tuning approach, as illustrated in Figure 4. The design is motivated by two key challenges: (1) the scarcity of large-scale, high-quality temporal annotations for future scene understanding; and (2) the need to align temporally distributed events based on partial visual context.

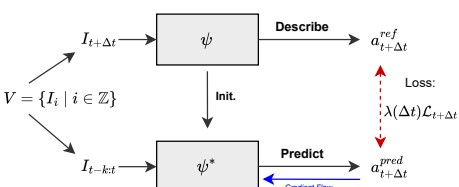

Instead of relying on expensive manual labels, we leverage the original pretrained model $\psi$ to generate pseudo reference descriptions $a_{t+\Delta t}^{ref}$ using ground-truth future frames $I_{t+\Delta t}$. We then fine-tune a new model $\psi^*$, initialized from $\psi$, to predict these descriptions from past-only inputs $I_{t-k:t}$, without access to future frames. This encourages the model not only to interpret the current visual input but also to imagine and temporally align possible future events. In addition, we incorporate a temporal **Chain-of-Thought (CoT)** formulation (Wei et al., 2022), where the model is guided to articulate intermediate reasoning steps describing how the scene evolves from the near future toward the further fu-

Figure 4: Proposed self-supervised approach to align temporal events and minimize incorrect or contradictory reasoning. Given a video sequence $V$, we generate detailed descriptions using a pretrained VLM $\psi$ as pseudo reference labels $a_{t+\Delta t}^{ref}$. We then fine-tune the model $\psi^*$, initialized from $\psi$, using only past frames as input and training it to predict descriptions of unseen future frames $a_{t+\Delta t}^{pred}$. A weighting function $\lambda(\Delta t)$ adjusts the contribution of each loss term based on the temporal distance $\Delta t$.

ture. This provides an auxiliary structural prior that encourages the model to reason through short-term transitions before imagining longer-horizon outcomes, leading to more stable and temporally coherent predictions. A time-aware weighting function $\lambda(\Delta t)$ is applied to modulate the loss contribution from different future steps, allowing the model to focus differently on short-term versus long-term temporal reasoning. In practice, we set $k = 5$, using 5 seconds of past observations (sampled at 1 frame per second) as input. We observed that increasing the window to 10 seconds did not improve performance but significantly increased computational cost. The weighting function $\lambda(\Delta t)$ is implemented as an exponential decay: $\lambda(\Delta t) = 2^{-\Delta t}$, assigning lower importance to predictions further into the future while still allowing for multi-scale temporal supervision. See Appendix B for more implementation details.

| VLM | Evaluation Method | | $S - M \downarrow$ | $M/S \uparrow$ |
|---|---|---|---|---|
| | Single-Trial ↑ | Multi-Trial ↑ | | |
| GPT-4o (Hurst et al., 2024) | 76.2% | 66.1% | 11.1% | 86.7% |
| GPT-4o-mini (Hurst et al., 2024) | 66.9% | 54.5% | 12.4% | 81.5% |
| LLV-v1.5-7b (Liu et al., 2023b) | 55.1% | 33.8% | 21.3% | 61.3% |
| LLV-v1.5-13b (Liu et al., 2023b) | 61.0% | 42.3% | 18.7% | 69.3% |
| LLV-Next-13b (Liu et al., 2024) | 41.8% | 18.7% | 23.1% | 44.7% |
| LLV-Video (Zhang et al., 2024) | 65.4% | 58.1% | **7.3%** | 88.8% |
| Qwen-VL-7b (Bai et al., 2023) | 24.6% | 4.6% | 20.0% | 18.7% |
| Qwen2.5-VL-7b (Bai et al., 2025) | 79.1% | 69.1% | 10.0% | 87.4% |
| CogVLM-17b (Wang et al., 2023) | 53.1% | 29.3% | 23.8% | 44.8% |
| Yi-VL-34b (Young et al., 2024) | 60.9% | 41.2% | 19.7% | 67.7% |
| Vid-LMA2 (Zhang et al., 2023a) | 67.6% | 54.3% | 13.3% | 80.3% |
| Baseline$^\dagger$ | 64.5% | 51.4% | 13.1% | 79.7% |
| FutureAgent$^\dagger$ | 62.7% | 52.1% | 10.6% | 83.1% |
| Baseline$^*$ | 73.5% | 63.5% | 10.5% | 85.7% |
| FutureAgent$^*$ | 72.3% | 64.0% | 7.8% | **89.2%** |

Table 1: In this evaluation we examine the ability of different VLMs on our evaluation dataset, where multiple answer options are shuffled across several rounds of answering by the VLMs. The accuracy change reflects the difference in performance between single-trial approach and multiple-trial answering, where the LLM must consistently identify the correct option in every round. This method minimizes the influence of random guessing by ensuring that only consistently correct answers are counted. $S - M$ denotes the performance drop from single-trial to multi-trial. The ratio $M/S$ represents the remaining performance.

## 5 EXPERIMENT AND ANALYSIS

In this section, we evaluate how well VLMs can reason about and describe potential future scenes based on preceding visual observations. Specifically, we analyze two key aspects: (1) whether the model can generate consistent responses under minimal input perturbations, which serves as an indicator of genuine understanding versus random guessing; and (2) whether the model can accurately reason about future scenes by interpreting the given history frames

### 5.1 EVALUATION SETUP AND IMPLEMENTATION DETAILS

All experiments were conducted on a server equipped with 4×A100-80GB GPUs. For fine-tuning, we utilized all 4 GPUs, while evaluation was performed using a single GPU for all models. In the FutureVQA benchmark, each input consists of a 5-second video segment, and the task is to reason about the future scene at time steps $t = 1$ to $t = 12$ seconds. For our fine-tuning method, we sampled training data from the OpenDV-YouTube dataset (Yang et al., 2024a), covering 16 cities across different continents. This subset comprises approximately 84k frames, each with a resolution of 1280×720, captured at various times of day. Training required approximately 140 GPU hours. Our base model uses Hermes-Yi-34B as the language backbone and CLIP-L (Radford et al., 2021) as the visual token encoder. It is pretrained using the LLaVA v1.6 (Liu et al., 2024) pipeline, and we refer to this model as Baseline$^*$ in our experiments. The fine-tuned version is denoted as Ours$^*$. We also evaluate a variant using Qwen-VL-32B as the language model, denoted as Baseline$^\dagger$ and Ours$^\dagger$ after fine-tuning.

### 5.2 CONSISTENCY AND RELIABILITY OF VLMS RESPONSE

In Table 1, we evaluate the performance of various VLMs on our proposed FutureVQA benchmark using the corresponding image for each question-answer pair as input—i.e., no future prediction is required. This setup serves both as a baseline for future scene reasoning and as a diagnostic to assess the consistency and reliability of VLM responses. Notably, we observe that all tested VLMs exhibit a significant drop in accuracy when the answer options are simply shuffled, despite the semantic content of the questions remaining unchanged. The most substantial decline occurs with CogVLM (Wang et al., 2023), which drops by 23.8%, followed by LLaVA-NeXT 13B (Liu et al., 2024) with a 23.1% decrease.

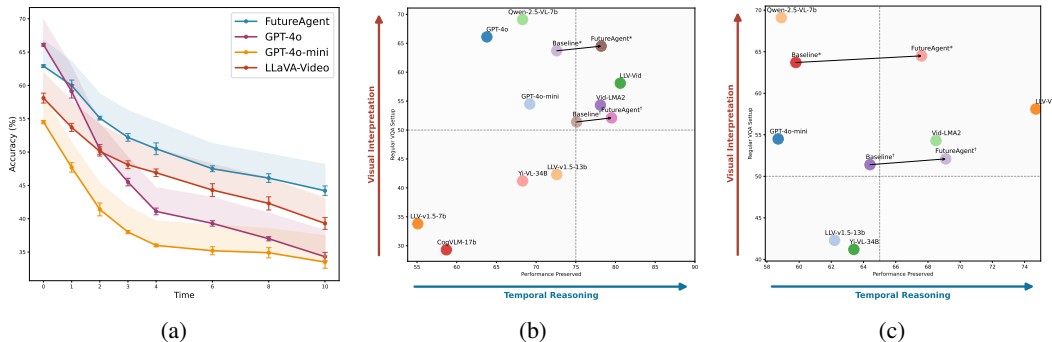

(a)             (b)             (c)

Figure 5: Temporal performance decay analysis on the FutureVQA dataset. (a) Accuracy decay across horizons, where solid lines denote four trials and shaded regions indicate fewer trials (1–3). (b) Relationship between regular VQA performance (y-axis) and relative long-horizon preservation (x-axis: Acc@12 divided by regular VQA accuracy). (c) Relationship between regular VQA performance (y-axis) and relative mean preservation (x-axis: $mAcc_{(1\to12s)}$ divided by regular VQA accuracy). Together, these plots show how well models retain their performance when extending from immediate perception to future prediction.

| Model | Accuracy ↑ | | | | | NDR ↓ | mRAR ↑ |
|---|---|---|---|---|---|---|---|
| | Acc@1s | Acc@4s | Acc@12s | $\Delta Acc_{1s}^{12s}$ | $mAcc_{(1\to12s)}$ | | |
| GPT-4o | 59.1% | 41.1% | 31.6% | -27.5% | 42.2% | 0.42 | 0.64 |
| GPT-4o-mini | 47.7% | 36.0% | 32.0% | -15.7% | 37.7% | 0.29 | 0.69 |
| LLV-v1.5-7b | 24.0% | 18.1% | 16.0% | **-8.0%** | 18.6% | 0.24 | 0.55 |
| LLV-v1.5-13b | 37.8% | 30.9% | 26.3% | -11.5% | 30.7% | 0.27 | 0.73 |
| LLV-Next-13b | 15.4% | 9.3% | 4.2% | -11.2% | 7.3% | 0.60 | 0.39 |
| LLV-Video | 53.7% | 46.5% | 43.4% | -10.3% | 46.8% | **0.18** | **0.81** |
| Qwen2.5-VL-7b | **61.9%** | 49.5% | 40.7% | -21.2% | 47.2% | 0.31 | 0.68 |
| CogVLM-17b | 22.8% | 19.4% | 14.0% | -8.8% | 17.2% | 0.30 | 0.59 |
| Yi-VL-34b | 38.1% | 30.0% | 26.1% | -12.0% | 28.4% | 0.29 | 0.70 |
| Vid-LMA2 | 52.4% | 41.2% | 37.2% | -15.2% | 42.4% | 0.28 | 0.78 |
| Baseline[†] | 49.8% | 44.1% | 33.1% | -16.7% | 38.6% | 0.33 | 0.75 |
| FutureAgent[†] | 49.2% | 46.7% | 36.0% | -13.2% | 41.4% | 0.25 | 0.79 |
| Baseline[*] | 60.2% | 48.2% | 38.1% | -22.7% | 46.1% | 0.36 | 0.73 |
| FutureAgent[*] | 60.8% | **50.7%** | **43.6%** | -16.6% | **50.1%** | 0.21 | 0.78 |
| w/o CoT | 60.5% | 48.4% | 41.3% | -19.2% | 48.2% | 0.30 | 0.75 |

Table 2: Accuracy (Acc) of models on our VQA benchmark at different future time frames. All accuracy values are evaluated across multiple trials to minimize the influence of random chance. The result suggest that models like GPT-4o, while showing strong ability in visual understdaning, fail to maintain consistent future scene reasoning across different time interval. [†][*]Our model is not trained with explicit temporal (video) label.

**Prompt-perturbation sensitivity vs. random guessing.** The performance drop in Table 1 across all models is largely attributable to random guessing, as the decrease scales with the number of options (four in our setup) and the number of trials. In contrast, Figure 5a shows error bars that reflect much smaller shifts when repeating four trials multiple times. These fluctuations (typically 0.5–1.2 points) arise from prompt-perturbation sensitivity: responses are inconsistent across trials but still exhibit a clear preference toward certain answers, rather than uniform randomness.

## 5.3 CAN VLMS "SEE" THE FUTURE?

Effective decision-making in dynamic environments should be grounded in accurate predictions. Here, we investigate whether VLMs are capable of reasoning about future scenes based on their interpretation of present visual cues, and whether they understand how events unfold over time. As shown in Table 2, we evaluate VLMs on our FutureVQA benchmark by asking them to answer questions about unseen future scenes using only the past five seconds of visual input. The task

challenges models to make predictions ranging from 1 to 12 seconds into the future. Each question is evaluated using a multi-trial protocol. Interestingly, we find that models that perform best in standard visual understanding tasks do not necessarily excel in future reasoning. For example, while GPT-4o demonstrates strong visual comprehension, its performance drop over time, measured by both $\Delta\text{Acc}_{1s}^{12s}$ and NDR, is significantly higher than that of other models. This suggests that, while equipped with strong visual interpretation capabilities, these models often fail to reason about how a scene evolves over time. In particular, they may struggle to understand how present events influence future outcomes, even if they generate accurate responses based on the current image.

In Figure 5b and Figure 5c, we observe that very poor visual interpretation ability typically coincides with weak temporal reasoning—an expected outcome since reliable reasoning requires accurate perception as a foundation. However, models such as GPT-4o (Hurst et al., 2024) and Qwen-2.5 (Bai et al., 2025), despite strong visual interpretation, experience significant drops when asked to predict the future, suggesting that good perception alone does not guarantee reliable temporal reasoning.

In (Table 3, Table 4), we compare how closely the predicted future scene descriptions match the model's own descriptions when the actual future image is provided. Ideally, if the prediction is accurate, both descriptions should align, as if the model had seen the future scene. Our results show that, after applying the proposed training method, the predicted descriptions become significantly more accurate and consistent across all time intervals.

| Model | Mean Score$_{(0 \to 12s)}$ $\uparrow$ | | | | |
|---|---|---|---|---|---|
| | $mB3$ | $mB4$ | $mRL$ | $mC$ | $mM$ |
| Baseline[†] | 10.7 | 6.0 | 22.8 | 2.3 | 25.4 |
| FutureAgent[†] | 20.3 | 19.8 | 35.2 | 11.3 | 34.6 |
| Baseline[*] | 12.3 | 7.1 | 25.2 | 3.6 | 28.5 |
| FutureAgent[*] | **28.8** | **22.7** | **37.3** | **12.3** | **39.2** |
| w/o CoT | 25.9 | 20.4 | 35.9 | 11.1 | 38.3 |
| w/o self-sup. | 11.8 | 6.9 | 24.7 | 2.3 | 26.0 |

Table 3: We compare how well our proposed model describes future scenes as if it "sees" them. A higher value indicates greater similarity between the reference description and the predicted description. The mean score, $m$, is computed over discrete time steps $t \in \mathbb{Z}_{[1,12]}$ seconds. B3: BLEU-3, B4: BLEU-4, R-L: ROUGE-L, C: CIDEr, M: METEOR.

| Model | Score Over Time $\uparrow$ | | | | |
|---|---|---|---|---|---|
| | S@1s | S@2s | S@4s | S@8s | S@12s |
| LLV-v1.5-7b | 2.59 | 2.67 | 2.07 | 2.52 | 2.25 |
| LLV-v1.5-13b | 2.13 | 1.92 | 1.94 | 2.49 | 2.40 |
| LLV-Next-13b | 2.11 | 2.87 | 2.26 | 2.57 | 2.15 |
| Baseline[†] | 4.88 | 4.01 | 2.96 | 2.34 | 2.41 |
| FutureAgent[†] | 5.31 | 5.01 | 3.98 | 3.44 | .2.46 |
| Baseline[*] | 5.36 | 4.23 | 3.03 | 3.22 | 2.98 |
| FutureAgent[*] | **6.43** | **6.12** | **5.33** | **5.04** | **4.66** |
| w/o CoT | 5.84 | 5.44 | 4.33 | 4.18 | 3.92 |
| (w/o self-sup. | 3.72 | 3.96 | 3.01 | 3.19 | 3.04 |

Table 4: Model-based evaluation of predicted caption quality across various time frames using GPT-4o, with a specific focus on objective descriptions, such as the accuracy of object appearance and location within the image.

# 6 LIMITATION AND DISCUSSION

While our approach offers data efficiency and improved temporal reasoning, it also presents trade-offs. The self-supervised fine-tuning relies on the quality of the baseline model; its limitations may propagate through pseudo labels. Future work could explore alternative forms of supervision such as constructing high-quality, large-scale training data. Similarly, although CoT prompting enhances reasoning without additional training, its step-by-step nature increases inference time. This may be a concern in real-time settings. A promising direction is to distill multi-step reasoning into a single-step model for faster inference. Despite these challenges, our framework provides a practical and extensible foundation for enhancing temporal understanding in VLMs.

# 7 CONCLUSION

We investigated the foresight capabilities of VLMs and found that, despite strong visual understanding, they struggle with consistent future scene reasoning. To address this, we introduced the FutureVQA Benchmark, a human-annotated dataset designed to evaluate VLMs' perception and prediction across different time intervals. Our experiments demonstrate that conventional models fail to maintain consistency in future predictions, while our self-supervised training pipeline improves temporal reasoning without requiring annotated temporal data. Notably, our model outperforms video-based VLMs despite lacking explicit temporal supervision. These findings highlight the need for better integration of visual perception and temporal reasoning in VLMs.

## ETHICS STATEMENT

This work uses publicly available and properly licensed driving-scene datasets, and all human-annotated question–answer pairs in FutureVQA were collected with informed consent and contain no personally identifiable information. Our method, FutureAgent, is designed solely to study the temporal reliability and consistency of VLM reasoning in offline driving scenarios; it is not intended for real-world autonomous driving or safety-critical deployment. We clearly report the limitations of our benchmark and method, and we caution that model outputs should not be used directly for vehicle control or decision making. We support responsible AI research by releasing our dataset construction details, evaluation metrics, and methodological choices transparently, and by encouraging safe, rigorous, and ethical use of this benchmark for analyzing VLM behaviors rather than operational driving systems.

## REPRODUCIBILITY STATEMENT

We have taken several steps to support reproducibility. The complete details of our model architecture, training objectives, and self-supervised fine-tuning procedure are described in Section 5 with additional implementation details provided in Appendix B. The construction process of the FutureVQA benchmark, including data selection, annotation protocol, and preprocessing steps, is documented in Appendix A of the appendix. Evaluation scripts in the supplementary materials to facilitate full reproducibility of our results.

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

# Appendix

## A FUTUREVQA

### A.1 DATASET CREATION AND QUALITY CONTROL

Our dataset creation aims to provide a benchmark that address VLMs ability in consistant and accurate future reasoning with focus on diverse questions costomized based on different scene. To achieve this we utilize the annotation pipeline operate with both human and AI agent, which to efficiently create the QA.

**(1) Human Expert QA Generation and Quality Control:** To construct a human-like benchmark dataset with high diversity, we employed five expert annotators to manually generate question-answer pairs based on selected clips from OpenDV-YouTube dataset (Yang et al., 2024a), covering multiple cities with different weathers. Each QA pair was subsequently reviewed and verified by 1–2 annotators to ensure clarity, unambiguity, and answerability based on the given input. Although time-consuming, this process results in a more diverse and naturally phrased QA dataset compared to rule-based or template-driven approaches.

Compared to existing works in the driving domain (see Figure 2), such as nuScenes-QA (Qian et al., 2023) and DRAMA (Malla et al., 2023), which rely on rule-based methods, or OmniDrive (Wang et al., 2024a), which uses GPT-generated data to construct large-scale datasets, our benchmark prioritizes diversity and human-like reasoning. While DriveLM-ns (Sima et al., 2023) incorporates human annotations for prediction and planning tasks, it still follows a rigid and highly structured question format, regardless of the uniqueness of each video clip. As shown in Table 5, despite being smaller in overall size, our dataset provides over $4\times$ more unique questions, nearly $3\times$ larger vocabulary, and over $400\times$ higher type-token ratio (TTR). Notably, more than 95% of our questions appear fewer than 10 times. In contrast, DriveLM contains over 85% of questions repeated more than $10^2$ times, over 20% more than $10^3$ times, and over 2% more than $10^4$ times, without considering the uniqueness of differences in scene content.

**(2) AI Quality Control and Multi-option Generation:** To minimize typographical errors, we employ GPT-4o to review all QA pairs generated by human annotators and automatically correct any detected typos. Following this, GPT-4o is further used to generate plausible but incorrect answer options based on the ground-truth answers provided by annotators.

To ensure that the resulting multiple-choice questions remain unambiguous—with only one clearly correct answer—each generated QA pair is manually reviewed by human annotators. This final verification step guarantees the quality and clarity of the multi-option format in our dataset.

| Dataset | N. Ques. | N. Uniq. Ques. | Vocab. Size | TTR |
|---|---|---|---|---|
| DriveLM(Pred.) | 123k | 15 | 69 | $4.1 \times 10^{-5}$ |
| DriveLM(Pred.&Percep.) | 285k | 234 | 150 | $4.1 \times 10^{-5}$ |
| Ours | 2.7k | **969** | **433** | $\mathbf{1.8 \times 10^{-2}}$ |

Table 5: Comparison of question diversity between our dataset and DriveLM. **N. Ques.** denotes the total number of questions; **N. Uniq. Ques.** represents the number of unique questions after de-duplication; **Vocab. Size** is the number of distinct words used in the questions; and **TTR** (Type-Token Ratio) measures lexical diversity, computed as the ratio of unique words to total words. The results highlight its greater linguistic diversity and reduced reliance on fixed templates.

### A.2 EVALUATION PROTOCOL

To address the limitations of conventional statistical-based metrics, we adopted an option-based answer format for evaluation, where each question has predefined multiple-choice answers (e.g., A: Yellow). The models were required to provide the corresponding option label (e.g., A) as the answer. See Figure 6 for the prompt and Algorithm 2 for the multi-trials evaluation.

Interestingly, during our experiments, we observed that not all models consistently adhered to this strict answer format. Some models would output answers like A: Yellow or simply Yellow. To account for this, we relaxed the evaluation criteria to accept both answer formats as correct. However, models like Qwen-VL-7B (Bai et al., 2023) still struggled to follow the instructions and produced responses such as *"The answer is A"*, *"The answer is A: Yellow"*, or other variations. Since following instructions is an important part of the evaluation, we did not further relax this restriction, which resulted in lower accuracy for these models, as shown in Table 1.

| Benchmark | Task | T. Size | Cust. Q | Ans. Type | Mul. Trl. | Mul. C. | Pred. | T-Pred. |
|---|---|---|---|---|---|---|---|---|
| nuScenes-QA Qian et al. (2023) | Drive VQA | 83.3k** | ✗ | Mixed | ✗ | ✓ | ✗ | ✗ |
| BDD-X Kim et al. (2018) | Drive Action | 2.6k | - | Sentence | ✗ | ✓ | ✗ | ✗ |
| DRAMA Malla et al. (2023) | Drive VQA | 11.6k | ✗ | Mixed | ✗ | ✗ | ✗ | ✗ |
| Rank2Tell Sachdeva et al. (2024) | Drive VQA | - | ✗ | Mixed | ✗ | ✓ | ✗ | ✗ |
| OmniDrive Wang et al. (2024a) | Drive VQA | 24k† | ✓ | Sentence | ✗ | ✓ | ✓ | ✗ |
| DriveLM-nS Sima et al. (2023) | Drive VQA | 73k* | ✗ | Sentence | ✗ | ✓ | ✓ | ✗ |
| MMBench Liu et al. (2023d) | Gen. I. QA | 1.7k | ✓ | Options | ✓ | - | - | - |
| LngVidBench Wu et al. (2024) | Gen. V. QA | 5.3k | ✓ | Options | ✗ | - | - | - |
| Video-MME Fu et al. (2024) | Gen. V. QA | 2.7k | ✓ | Options | ✗ | - | - | - |
| MME Fu et al. (2023a) | Gen. I. QA | 2.1k | ✗ | Y/N | ✗ | - | - | - |
| Ours | Drive VQA | 2.8k | ✓ | Options | ✓ | ✓ | ✓ | ✓ |

Table 6: Comparison of existing VLM benchmarks. Key aspects of dataset creation include test size (T. Size), whether questions are customized for different scenarios and video clips (Cust. Q), answer type (Ans. Type), multi-trial evaluation for each question (Mul. Tri), inclusion of multiple cities (Mul. C.), presence of perception tasks (Perc.), inclusion of prediction tasks (Pred.), and whether the dataset challenges VLMs with time-specific prediction (T-Pred.). Our benchmark dataset consists of fully human-annotated QA pairs tailored to different scenes, rather than relying on rule-based methods. Furthermore, our dataset challenges VLMs to predict future scenes at specific time intervals, requiring precise temporal reasoning to differentiate between near-future and far-future events.

†: The QA pairs are fully generated by GPT-4. ** Fully rule-based (no human annotators), * Semi-rule-based labeling (with human annotators for certain tasks).

---

**FutureVQA Prompt**

Imagine you are looking at the image **{future_second}** second after the input frames and answer the following question:
Question: **{question}**
Options: **{options}**

Please choose the most appropriate answer from the given options. Respond with the option without any explanation, for example, if the answer is B: Yellow, your answer should be: B

Figure 6: The prompt used to instruct VLMs to predict the future scene and answer the corresponding question.

### A.3 VQA CATEGORY

To evaluate the diverse reasoning capabilities of VLMs, we classify VQA tasks into the following categories. These categories are not mutually exclusive, as a single question can belong to multiple categories depending on the type of reasoning required.

- **Hallucination:** This category evaluates the model's ability to avoid providing incorrect information about objects or features that do not exist in the scene. (e.g., *"How many blue cars do you see in this image?"*) Such questions are especially challenging when an object has just left the scene.

- **General:** General questions involve straightforward scene understanding or recognition tasks that do not require spatial or temporal reasoning. Examples include identifying landmarks, objects, or common scene elements (e.g.,

*"What is the landmark in the middle of the image?"*).

- **Traffic Understanding:** This category targets traffic-related reasoning, including understanding road signs, speed limits, or dynamic traffic scenarios. These questions often require knowledge specific to driving environments (e.g., *"What is the speed limit here?"*).

- **Absolute Location:** Absolute location questions focus on the spatial properties of objects in the scene, such as identifying specific positions or attributes relative to the image boundaries (e.g., *"What color is the car on the far right of the image?"*).

- **Relative Position:** Relative position questions require understanding the spatial relationships between multiple objects in the scene. These questions test the model's ability to interpret multiple objects interaction (e.g., *"Describe the vehicle in front of the taxi."*).

By introducing these categories, we aim to provide a comprehensive evaluation framework for VLMs, covering both basic scene understanding and complex reasoning tasks. See Figure 7 for the examples.

### A.4 ANALYSIS ON DIFFERENT FUTUREVQA CATEGORIES

To establish a baseline for expected performance in the FutureVQA, we analyze various VLMs

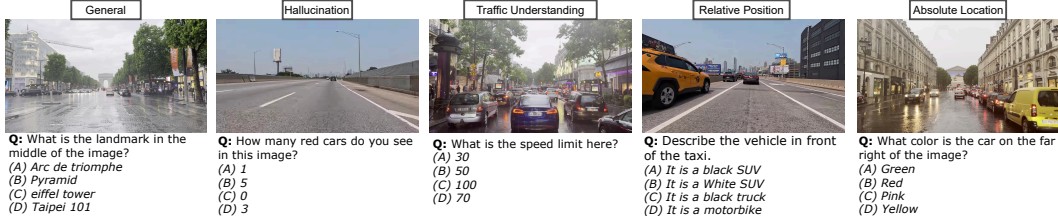

Figure 7: Examples of visual question answering (VQA) tasks categorized into different types: **Hallucination**, **General**, **Traffic Understanding**, **Absolute Location**, and **Relative Position**. Each question is categorized based on the type of reasoning it requires; however, a single question can belong to multiple categories simultaneously, depending on its context and the type of information needed.

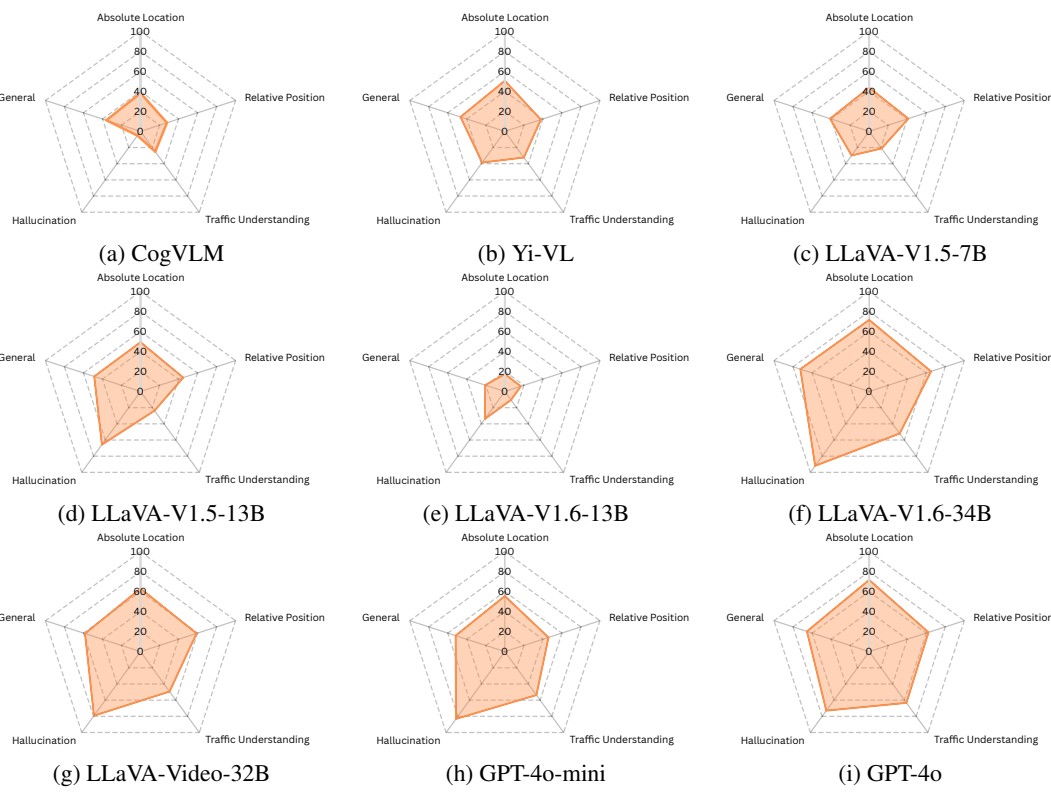

Figure 8: Radar plots comparing the performance of various models across five VQA categories: Hallucination, General, Traffic Understanding, Absolute Location, and Relative Position. In this experiment, models perform regular VQA on images, with the actual images provided as input. The plots illustrate the strengths and weaknesses of each model in handling different reasoning tasks, providing a comparative baseline for understanding the capabilities of existing VLMs before extending to future image QA tasks.

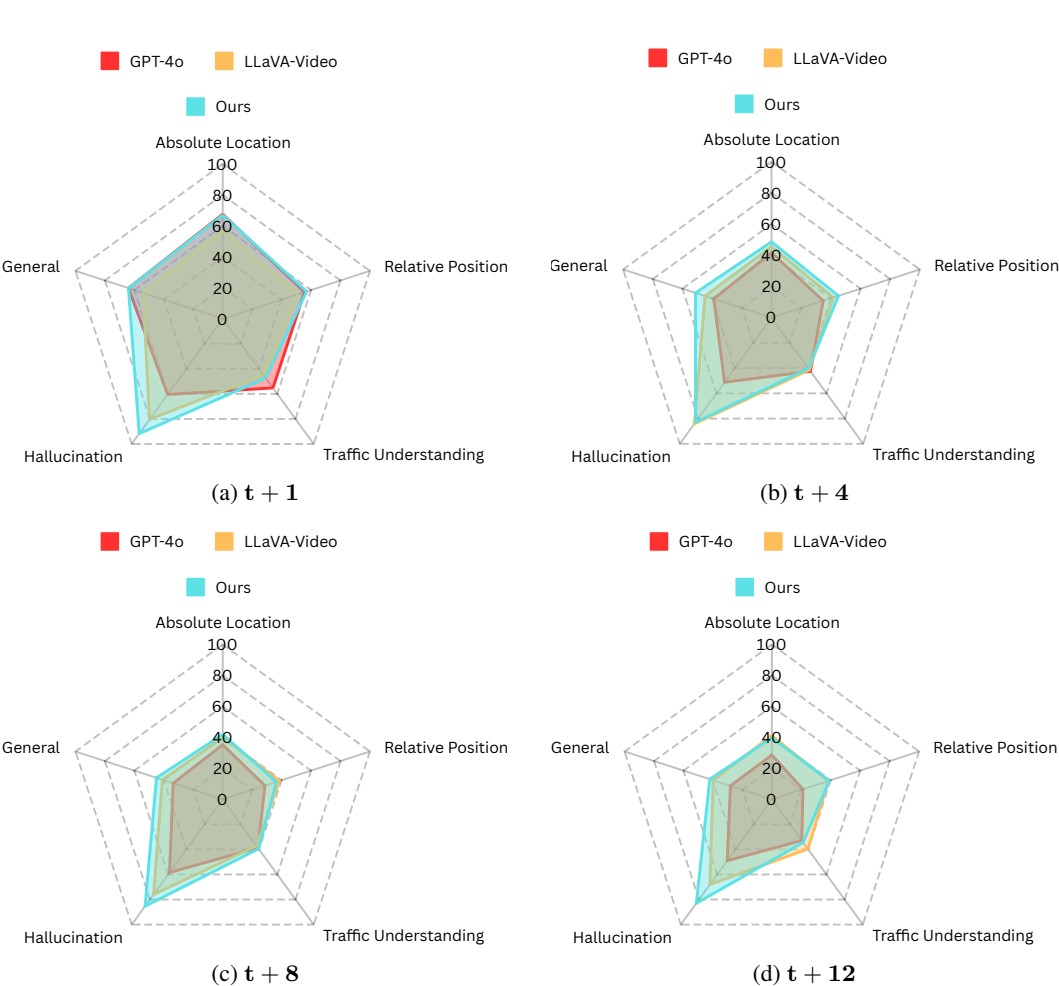

Figure 9: Radar plots comparing the performance of different models across various VQA categories (Hallucination, General, Traffic Understanding, Absolute Location, and Relative Position) at different future time steps: (a) $t + 1$, (b) $t + 4$, (c) $t + 8$, and (d) $t + 12$. The results highlight that while most models maintain robustness in hallucination detection, their performance in other categories, particularly traffic understanding and spatial reasoning, declines as the time offset increases.

on our benchmark dataset across different categories . In this baseline analysis, VLMs perform regular VQA, where the actual images corresponding to the questions are provided as input.

As shown in Figure 8, we evaluate models includes CogVLM (Wang et al., 2023), Yi-VL (Young et al., 2024), LLaVA series (Liu et al., 2023a; 2024) and GPT-4o, the results suggest that traffic understanding appears to be a relatively weak area for many existing VQA models. Most models do not exhibit significant differences in their capability to handle absolute or relative position questions. Additionally, for hallucination-related tasks, where models are asked about nonexistent objects, most models perform well when the image is provided, effectively avoiding incorrect predictions. These findings highlight the strengths and weaknesses of current VLMs and provide a foundation for evaluating their potential performance in future image QA tasks.

In Figure 9, we further compare the performance of VLMs across different question categories when asked to predict future scenes. As time progresses, we observe that GPT-4o's performance degrades significantly across all categories, with the most notable decline in questions related to relative and absolute positioning.

---

**Algorithm 3** Temporal Chain-of-Thought Future Scene Reasoning

---

**Require:** VLM $\psi$, Observed Frames $I_{t-5:t}$, Target Future Step $\Delta t$, Question $Q_{t+t_\Delta}$
1: $D_0 \leftarrow$ Initialize empty future description
2: **for** $i = 1$ to $\Delta t$ **do**
3: $\quad D_i \quad\quad\quad\quad\quad\quad\quad\quad\quad \leftarrow$ $\psi.\texttt{describe\_future}(I_{t-5:t}, D_{i-1}, i)$
4: **end for**
5: $ans \leftarrow \psi.\texttt{answer}(Q_{t+t_\Delta}, D_T)$
6: **return** $ans$

---

## B DETAIL IMPLEMENTATION

### B.1 CHAIN-OF-THOUGHT

To enhance temporal reasoning, we adopt a Chain-of-Thought (CoT) prompting strategy in which the VLM predicts the future scene progressively, one step at a time. Rather than directly predicting the outcome at a future timestamp, the model is encouraged to reason through each intermediate step—first predicting $t = 1$, then $t = 2$, and so on, until the final target time is reached, see Algorithm 3. At each step, the model uses the history frames along with its previous predictions to generate the next future scene description. This design mimics human-like sequential foresight and allows the model to build up an understanding of how the scene may evolve over time. For practical computational efficiency, we limit the maximum number of steps to 4. This step-wise reasoning not only improves temporal consistency but also provides interpretable intermediate predictions that make the model's reasoning process more transparent and grounded in scene dynamics.

### B.2 VISUAL INPUT ENCODING

**Memory Decay Sampling.** Our implementation of the memory decay sampler leverages a transformer-based framework with learnable sampling queries $Q = \{q_1, q_2, \ldots, q_n\}$, where $n$ is the total number of queries set as the initial number of tokens. These queries are initialized at the beginning of training and are optimized to extract temporal information relevant to the task. Let the current time be $t_0$, and let the number of tokens provided by the image encoder be $n_0$. The decay factor for the frame at time $t_0 - i$ is defined as $\left(\frac{1}{2}\right)^i$. Accordingly, the first $n_0 \cdot \left(\frac{1}{2}\right)^i$ queries are utilized in the cross-attention mechanism to represent the frame at $t_0 - i$.

**Adaptive Token Sampling.** In our implementation, frame similarity is evaluated by first com-

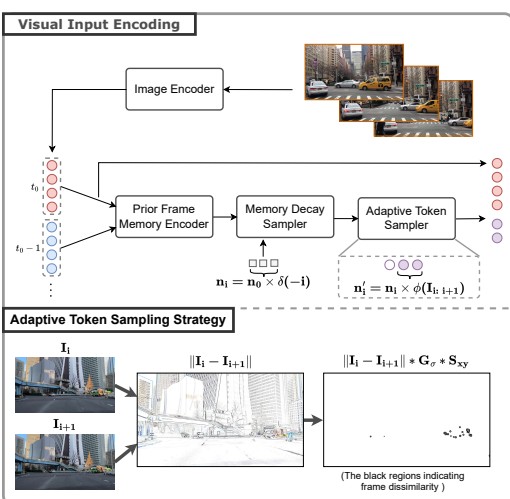

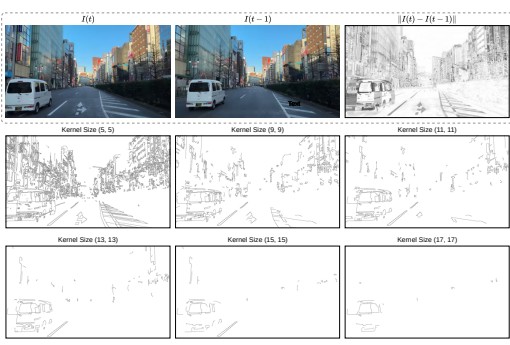

Figure 11: Visualization of frame similarity evaluation using Gaussian smoothing followed by the Sobel operator with different kernel sizes. The input images have a resolution of $1280 \times 720$ pixels. The difference between two consecutive frames, $|I(t) - I(t-1)|$, is computed, smoothed using Gaussian filters with kernel sizes of 5, 9, 11, 13, 15, and 17, and then processed with the Sobel operator, $S_{xy}$, to highlight changes. For better readability, the colors of the similarity maps are inverted.

Figure 10: Overview of our visual encoding pipeline. The goal is to minimize the number of tokens while maintaining similar performance. In the context of autonomous driving videos, recent frames typically have greater influence on upcoming events. To reflect this, the **Memory Decay Sampler** assigns fewer queries to older frames, while the **Adaptive Token Sampler** adjusts the number of tokens based on the similarity between adjacent frames. The **Prior Frame Memory Encoder** is a transformer-based module that integrates temporal information from preceding frames.

puting the difference between two consecutive frames, $|I(t) - I(t-1)|$. To reduce noise introduced by high-frequency details, such as windows on distant buildings in urban environments, a Gaussian filter, $G_\sigma$, is applied to smooth the difference map while preserving significant changes. Finally, a Sobel operator, $S_{xy}$, is used to highlight the structural changes between the frames.

During our experiments, we tested multiple Gaussian filter kernel sizes and determined that a kernel size of 13 strikes the best balance between reducing noise and preserving important structural details. The comparison is shown in Figure 11. After computing the similarity maps, we measure the amount of highlighted area and then scale and cap the values for consistency. On average, the scaling factor is approximately 0.5 across our evaluation dataset.

| Time | Model | Scores ↑ | | | | |
|------|-------|-----|-----|-----|-----|-----|
| | | B-3 | B-4 | R-L | C | M |
| +1s | Baseline* | 13.4 | 6.1 | 25.0 | 2.4 | 25.9 |
| | Ours* | **32.2** | **26.2** | **40.2** | **17.7** | **41.5** |
| | w/o CoT | 28.1 | 22.7 | 38.2 | 15.1 | 40.2 |
| | w/o self-sup. | 12.1 | 7.2 | 24.9 | 2.7 | 26.2 |
| | $t_{0s:-10s}$ | 32.5 | 26.5 | 40.0 | 17.6 | 41.3 |
| | w/o Adpt. Sam. | 32.3 | 26.2 | 40.4 | 17.3 | 41.6 |
| | w/o Mem. Sam. | 31.8 | 26.2 | 40.1 | 17.0 | 41.2 |
| +4s | Baseline* | 22.5 | 6.0 | 24.4 | 2.6 | 25.5 |
| | Ours* | **28.6** | **22.5** | **37.1** | **11.8** | **39.1** |
| | w/o CoT | 25.6 | 20.1 | 35.9 | 11.0 | 38.4 |
| | w/o self-sup. | 11.6 | 6.7 | 24.4 | 2.0 | 25.8 |
| | $t_{0s:-10s}$ | 32.1 | 26.2 | 40.7 | 17.7 | 41.5 |
| | w/o Adpt. Sam. | 32.7 | 26.3 | 40.6 | 18.0 | 41.5 |
| | w/o Mem. Sam. | 32.7 | 25.6 | 40.8 | 18.1 | 41.5 |
| +8s | Baseline* | 11.2 | 6.2 | 25.1 | 2.2 | 25.4 |
| | Ours* | **27.5** | **21.4** | **36.2** | **10.1** | **38.3** |
| | w/o CoT | 24.1 | 18.6 | 34.9 | 9.7 | 37.6 |
| | w/o self-sup. | 12.0 | 7.1 | 24.9 | 2.3 | 26.3 |
| | $t_{0s:-10s}$ | 32.2 | 25.8 | 40.2 | 17.7 | 41.5 |
| | w/o Adpt. Sam. | 32.3 | 26.6 | 41.5 | 17.5 | 41.5 |
| | w/o Mem. Sam. | 32.0 | 26.4 | 41.9 | 17.1 | 41.5 |
| +12s | Baseline* | 11.3 | 7.2 | 23.9 | 2.2 | 25.5 |
| | Ours* | **26.7** | **20.6** | **35.6** | **9.4** | **37.7** |
| | w/o CoT | 25.6 | 20.1 | 34.4 | 8.6 | 36.9 |
| | w/o self-sup. | 11.5 | 6.7 | 24.4 | 2.0 | 25.8 |
| | $t_{0s:-10s}$ | 31.2 | 26.0 | 39.5 | 17.0 | 41.5 |
| | w/o Adpt. Sam. | 32.0 | 25.5 | 39.6 | 16.9 | 41.5 |
| | w/o Mem. Sam. | 32.1 | 26.2 | 40.4 | 17.9 | 41.5 |

Table 7: In this comparison the reference captions are from regular image captioning, while the compared captions are generated by our fine-tuned model which perform future scenes captioning with only previous frames are given. B-3: BLEU-3, B-4: BLEU-4, R-L: ROUGE-L, C: CIDEr, M: METEOR.

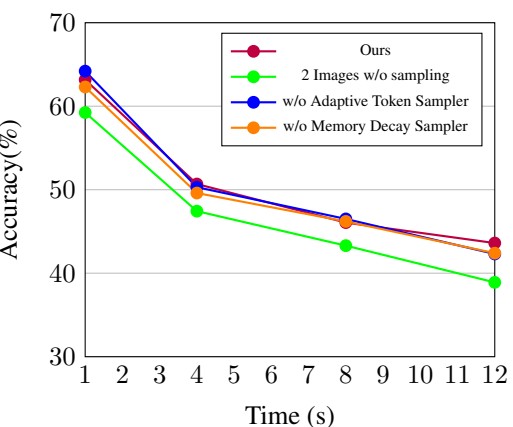

Figure 12: Accuracy over time in the FutureVQA task with different visual input encoding strategies, showing that our sampling approach reduces the number of required tokens while maintaining higher performance.

## C   ADDITIONAL EVALUATION

### C.1   ABLATION STUDY ON SAMPLING STRATEGY

Our choice of the number of visual input frames is guided by two main considerations: (1) performance and (2) hardware constraints. The objective is to minimize the number of visual tokens while maintaining competitive performance. Detailed results at specific time steps are provided in Table 7 and Figure 12.

We observe that extending the input range from the past 5 seconds to the past 10 seconds does not lead to significant performance gains, yet results in increased computational cost. On the other hand, reducing the input to only the past 2 seconds leads to a slight drop in performance.

Similarly, ablating either the memory decay sampler or the adaptive token sampler individually does not substantially affect the final accuracy, while reducing visual token usage by approximately 75%. This highlights the efficiency of our sampling strategy in balancing performance and computational cost.

### C.2   HALLUCINATION AND MODE COLLAPSE

Generating accurate captions is the first and most crucial step in our training methodology. We experimented with various models for this task; however, we observed that not all models are capable of providing objective and accurate cap-

tions that comprehensively describe all elements in the scene. For instance, models like LLaVA-V1.5-7B (Liu et al., 2023b) tend to generate repetitive sentences and often produce hallucinations, exaggerating or inaccurately inferring details that are not present in the image. Figure 13 illustrates examples of these issues, showcasing captions that overstate the number of objects in the scene and use overly similar and redundant phrasing. These limitations highlight the need for more robust captioning models to ensure high-quality data generation for downstream tasks.

## D   PROMPT TEMPLATE

In this section, we describe the unified prompt template used for our experiments across three key tasks: captioning evaluation, regular VQA, and FutureVQA. The template, shown in Figure 14, standardizes the model's input format to ensure consistent and fair evaluation.

For captioning evaluation, the model generates captions for a given image, which are subsequently scored by GPT-4o acting as a judge. GPT-4o is instructed to provide a score between 1 and 10 based on the objective aspects of the caption, explicitly disregarding subjective elements such as mood or atmosphere.

In the regular VQA task, the model is provided with the input image and a set of predefined multiple-choice options. It is required to select the most appropriate answer, establishing a baseline for evaluating the model's performance when the image is explicitly available. In Figure 15 and Figure 16, we show that even without explicitly tuning the baseline model, having sufficient knowledge to interpret the visual input does not translate into temporal reasoning ability. The model fails to understand how events unfold over time and cannot align the scene for both near and distant future predictions.

Figure 13: Comparison of captions generated by LLaVA-V1.5-7B (Liu et al., 2023b) and LLaVA-V1.6-34B (Liu et al., 2024). While LLaVA-V1.5-7B produces shorter and repetitive captions with occasional hallucination, LLaVA-V1.6-34B generates significantly longer and more detailed descriptions. Additionally, LLaVA-V1.6-34B exhibits a varied response pattern, providing distinct levels of detail and focus when presented with different images.

**GPT-4o as Judge for Captioning Evaluation**

Please act as an impartial judge and evaluate the quality of the image caption provided by an AI assistant displayed below. Your evaluation should specifically assess the accuracy of object presence and positioning within the image, disregarding any subjective descriptions like vibe, atmosphere, or general impressions. Focus solely on whether the caption correctly reflects the precise positioning and presence of each object mentioned. Begin your evaluation by providing a short explanation. Be as objective as possible. After providing your brief explanation, please rate the response on a scale of 1 to 10 by strictly following this format: '[[rating]]', for example: 'Rating: [[5]]'.
Caption by the AI assistant: *{caption}*

**Regular VQA**

Answer the following question based on the image:
Question: *{question}*
Options: *{options}*
Please choose the most appropriate answer from the given options. Respond with the option without any explanation, for example, if the answer is B: Yellow, your answer should be: B

Figure 14: Prompts used for three tasks: GPT-4o as a judge in captioning evaluation and Regular VQA on our annotated evaluation dataset. Each prompt is tailored to the specific requirements of its respective task.

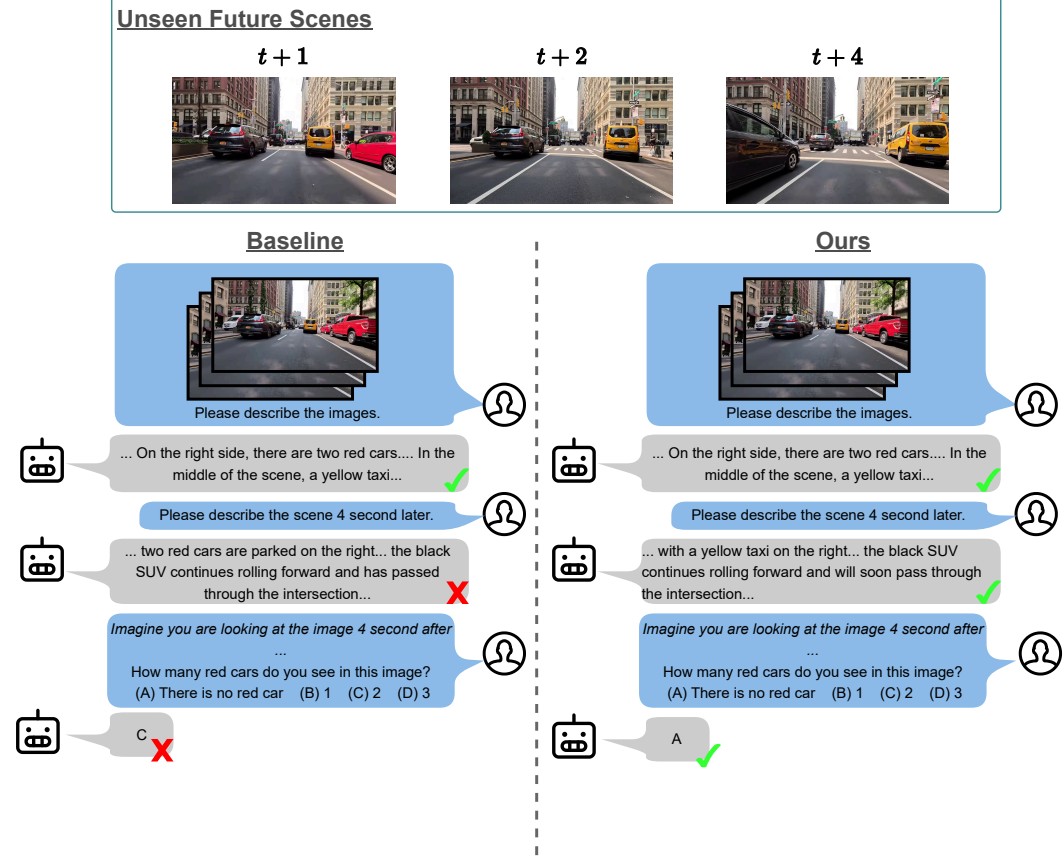

Figure 15: Quantitative result.

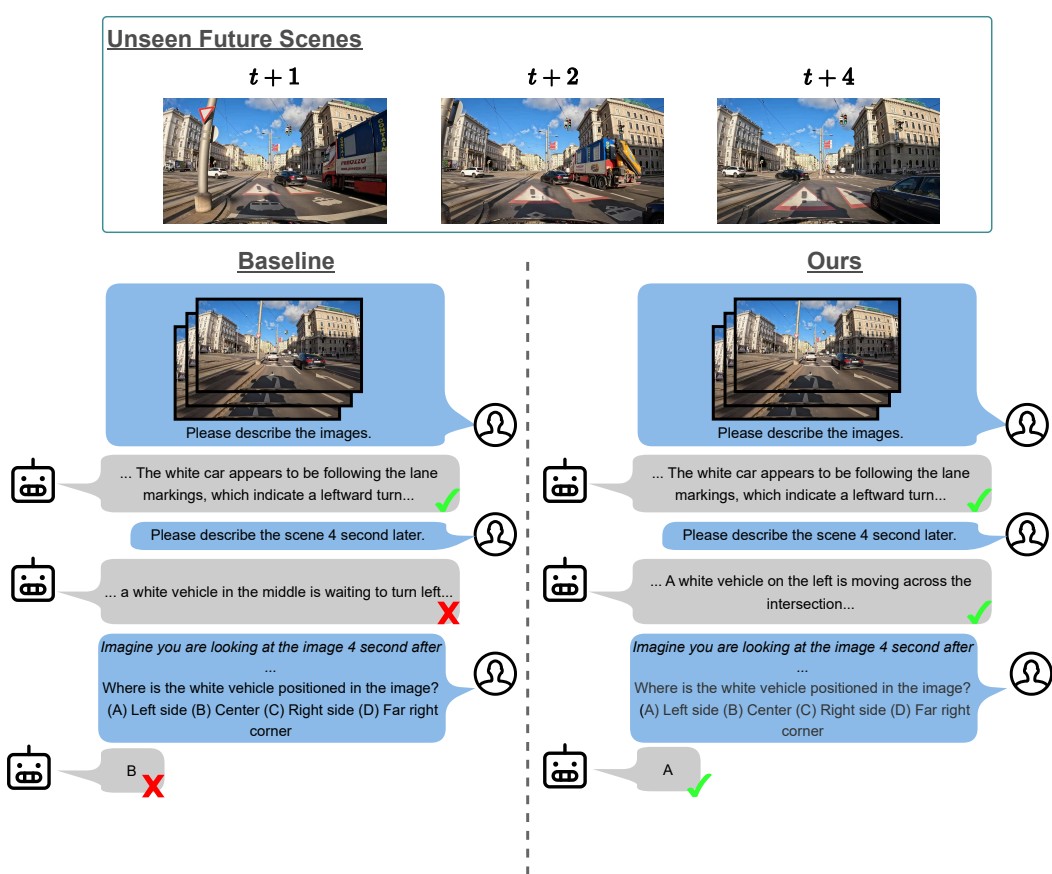

Figure 16: Quantitative result.

