# OpenReview forum: "The Challenge of Reliable Vision–Language Model Responses in Driving"
_ICLR.cc/2026/Conference — Submitted to ICLR 2026_

### Official Review · Reviewer_HYS6 · 2025-10-26

**Soundness:** 3
**Presentation:** 3
**Contribution:** 3
**Rating:** 4
**Confidence:** 3

**Summary:**

The paper probes whether VLMs used as driving assistants truly perform temporally grounded reasoning, finding two reliability failures—response inconsistency under semantics-preserving perturbations and weak temporal reasoning.

In addition, It introduces FutureVQA and an evaluation protocol (self-aligned future descriptions + multi-trial consistency) to test future-scene reasoning over 1–12-second horizons.

**Strengths:**

* The paper formalizes reliable temporal reasoning with explicit alignment between past-only and future-conditioned predictions and gives concrete measures under semantics-preserving perturbations.
* The paper propose a well-constructed dataset that targets future reasonin. Human-annotated FutureVQA focus on the time-specific prediction with diverse, naturally phrased questions and a multi-trial protocol.

**Weaknesses:**

* The main concern is the scale and context limitations. The benchmark contains 2.7k human-annotated QA and each input provides only a 5-second history while evaluating up to 12 s, which may underrepresent longer-horizon dynamics and diverse real-world conditions.
* Evaluation may be judge-biased. Future caption quality is partly scored by a single model-based judge (GPT-4o), and text similarity metrics (BLEU/ROUGE/CIDEr) are used—both may poorly capture safety-critical temporal reasoning and can introduce evaluator bias.

**Questions:**

The primary concern is the limitation in scale and context. Providing additional clarification here would be helpful.

---

> ### Author Response · Authors · 2025-11-22
> **Author Response (1/1)**
>
> We thank the reviewer for highlighting these important concerns regarding dataset scale, context length, and evaluation bias. We address each of these points in detail below.
>
> ### Weakness 1
>
> > "The primary concern is the limitation in scale and context. Providing additional clarification here would be helpful."
>
> **2.7k human-annotated QA:**
>
> As shown in **Section A, Table 6**, the scale of our 2.7k human-annotated QA pairs is comparable to many existing human annotated benchmarks (MMBench, Video-MME etc.).
>
> While other AI-generated datasets can achieve much larger scale.  Several studies have also raised concerns about the “pollution” effect of synthetic data, which may hinder progress if used without careful control. Even though they focus on training data, a similar concern applies to evaluation: high-stakes benchmarks such as MMLU and AIME, which are used to compare models like ChatGPT and DeepSeek, are deliberately manually constructed to avoid such biases.
>
> Thus, the additional effort required to ensure data quality, and the importance of keeping the benchmark genuinely human-authored, should be considered when evaluating dataset scale.
>
> Additionally, in **section A table 5** we also demonstrate the **diversity** of our proposed dataset even with a smaller scale.
>
> **5-Seconds history:**
>
> We would like to clarify that while each question in FutureVQA uses a 5-second history as input, the video clips used for question generation include at least a **10-second temporal buffer**, ensuring sufficient contextual coverage for each evaluated segment. To further examine the impact of context length, we conducted additional experiments varying the number of history frames (see **Table 7, Figure 12** and the results below). While incorporating longer history windows can provide slightly richer temporal cues, we observed that beyond **5 seconds**, models show **no significant improvement** in temporal reasoning accuracy, while computational cost increases notably. This finding suggests that in driving scenes, where the environment evolves rapidly, the most recent observations have the strongest influence on future dynamics.
>
> For this reason, we adopted the 5-second context as a balanced design choice in the experiment, and it is also possible to extend it to 10-seconds history context.
>
> | Model | 5 sec (mAcc) | 10 sec (mAcc) |
> | --- | --- | --- |
> | Ours | 50.1% | 49.8% |
> | GPT-4o | 42.2% | 42.0% |
> | LLv-Video | 46.8% | 46.9% |
>
> ### Weakness 2
>
> > "Evaluation may be judge-biased. Future caption quality is partly scored by a single model-based judge (GPT-4o), and text similarity metrics (BLEU/ROUGE/CIDEr) are used—both may poorly capture safety-critical temporal reasoning and can introduce evaluator bias."
>
> 1. We appreciate the reviewer’s thoughtful observation regarding evaluator bias.
>
>     We agree that reliably measuring VLM outputs, especially for safety-critical temporal reasoning, is inherently challenging and remains an open research problem [10,11].
>
>     To mitigate the limitations of any single evaluation method, our work intentionally combines **multiple complementary metrics**, including:
>
>     - **model-based judging** (GPT-4o),
>     - **text-similarity metrics** (BLEU, ROUGE, CIDEr, METEOR),
>     - **consistency-focused metrics** (multi-trial v.s. single trial, S-M, S/M),
>     - **temporal decay metrics** (Acc@t, mAcc, NDR, mRAR).
>
>     Our goal is not to claim that any single metric is perfect, but rather to construct a **comprehensive and cross-validated evaluation protocol** that captures different aspects of temporal alignment, consistency, and prediction reliability.
>
>     We view the design of more robust and unbiased evaluation tools for VLMs as an important direction for future work, and we appreciate the reviewer for highlighting this important challenge.
>
> [10] Li, Haitao, et al. "Llms-as-judges: a comprehensive survey on llm-based evaluation methods." *arXiv preprint arXiv:2412.05579* (2024).
>
> [11] Chang, Chun-Peng, Alain Pagani, and Didier Stricker. "3d spatial understanding in mllms: Disambiguation and evaluation." *2025 IEEE International Conference on Robotics and Automation (ICRA)*. IEEE, 2025.

---

### Official Review · Reviewer_Z1qg · 2025-10-29

**Soundness:** 2
**Presentation:** 2
**Contribution:** 2
**Rating:** 4
**Confidence:** 4

**Summary:**

This paper investigates VLM reliability in driving, finding models struggle with response inconsistency and temporal reasoning. It introduces FutureVQA, a human-annotated benchmark for future prediction. It also proposes FutureAgent, a self-supervised method that trains the model to predict pseudo-descriptions of future frames, improving temporal consistency.

**Strengths:**

This paper shifts focus from simple accuracy to the critical issues of reliability and temporal reasoning in driving. The analysis of response inconsistency using option shuffling is a simple and effective diagnostic. The introduction of FutureVQA provides a valuable, human-annotated resource for the field. The paper is clearly written, and the proposed FutureAgent method is an intuitive self-supervised approach that demonstrates improved performance. This work provides a useful framework for evaluating VLM foresight.

**Weaknesses:**

A key limitation lies in the problem's formulation. The FutureAgent task trains the model to predict the single, recorded future from the dataset. This setup treats the future as a passive, deterministic event. However, for a reliable driving assistant, the future is conditional on the ego-vehicle's own actions (e.g., braking vs. accelerating). The current method trains for passive prediction of what did happen, not for the action-conditional foresight of what might happen given different choices. This overlooks the agent's own influence on the environment.

**Questions:**

1. The FutureAgent task trains the model to passively predict a single, recorded future. Do the authors agree this is a limitation? How might the proposed method be extended to learn action-conditional future reasoning (e.g., "What will happen if I brake now?" vs. "...if I continue at this speed?")?
2. Table 1 shows that FutureAgent reduces the accuracy drop (the "S-M" column) compared to its baseline. Could you elaborate on why you believe this specific self-supervised task improves this measure of consistency?
3. The exponential decay weighting prioritizes short-term predictions. Have you experimented with other weighting functions, such as one that gives more weight to challenging long-term predictions?
4. How do you interpret the performance of FutureAgent compared to models explicitly trained on video dataset?

---

> ### Author Response · Authors · 2025-11-22
> **Author Response (1/2)**
>
> ### Question 1
> > "The FutureAgent task trains the model to passively predict a single, recorded future. Do the authors agree this is a limitation?"
>
> We appreciate the reviewer’s insightful comment regarding **action-conditioned future reasoning**, and we agree that it represents an important direction for future research, especially in closed-loop simulation and edge case analy, where prediction and planning are jointly considered.[7,8,9]
>
> We would like to clarify that our work focuses on evaluating and enhancing the **reliability** of VLM responses when interpreting the surrounding scene. Our evaluation follows the common setup used in many prediction and forecasting works, where interactions between vehicles are assumed to be constrained by traffic rules and typical driving behavior. Under these constrain, one can reasonably infer the future scene dynamics without explicitly observing the current control actions of the ego and/or surrounding vehicles. For example, when a vehicle is in a right-turn-only lane, all safe drivers are expected to turn right.
>
> Therefore, our task targets a foundational **reliability problem** for VLMs in driving, which **does not conflict with the reviewer’s proposed action-conditioned foresight**. We envision that once a VLM can provide consistent and temporally aligned reasoning, developing an action-conditioned VLM would be a valuable and promising next step.
>
> A possible extension is to condition the model on high-level action cues, enabling richer causal predictions;  The main challenge might be in acquiring datasets with reliable action annotations or counterfactual trajectories to supervise such reasoning.
>
> ### Question 2
> > "Table 1 shows that FutureAgent reduces the accuracy drop (the "S-M" column) compared to its baseline. Could you elaborate on why you believe this specific self-supervised task improves this measure of consistency?"
>
> While our self-supervised tuning is primarily designed to strengthen temporal reasoning and align events across time, we also incorporate Chain-of-Thought (CoT) reasoning (**Section 4, Appendix B**). When reasoning about future scenes, the model is prompted to first infer the *near* future before extending its reasoning to more distant states. In **Section 5, Table 1**, the model is instructed to think about the near future before answering the question; this additional reasoning step acts as a “hint,” and empirical results show that the model produces more consistent answers. Notably, after applying our tuning and CoT, **single-trial accuracy decreases while multi-trial accuracy increases**, suggesting that the model becomes more stable and less sensitive to prompt perturbations.
>
> In conclusion, while self-supervision improves temporal alignment and strengthens temporal reasoning, we find that the primary improvement in mitigating inconsistency caused by **prompt-perturbation sensitivity** comes from the use of Chain-of-Thought. Our ablation study further supports this observation: without CoT, the performance gain is minimal, less than **0.2%**.

---

> ### Author Response · Authors · 2025-11-22
> **Author Response (2/2)**
>
> ### Question 3
>
> > "The exponential decay weighting prioritizes short-term predictions. Have you experimented with other weighting functions, such as one that gives more weight to challenging long-term predictions?"
>
> This is indeed an insightful question and one that we also explored in the early stages of our training design. Intuitively, since long-term predictions are more challenging, one might expect that assigning them larger weights could encourage the model to learn these difficult cases better. However, our empirical results showed that this strategy led to significantly worse performance.
>
> Our interpretation is that the distant future carries inherently high uncertainty, and emphasizing these noisy, hard-to-predict targets makes the optimization problem overly difficult. In practice, we observed **unstable training dynamics** and behavior reminiscent of **mode collapse**, where the model converged to degenerate predictions.
>
> In contrast, prioritizing short-term predictions through exponential decay provides a more reliable supervision signal, resulting in more stable training and overall better performance.
>
> | Model | Acc@1 | Acc@4 | Acc@12 | $\Delta Acc^{12s}_{1s}$ | mAcc |
> | --- | --- | --- | --- | --- | --- |
> | Decay Weighting | 60.8 | 50.7 | 43.6 | -16.6 | 50.1 |
> | Increase Weighting | 38.4 | 31.6 | 25.5 | -12.9 | 27.6 |
> | Uniform Weighting | 58.3 | 47.3 | 39.9 | -18.4 | 46.1 |
>
> ### Question 4
>
> > "How do you interpret the performance of FutureAgent compared to models explicitly trained on video dataset?"
>
> In our experiments, we observe that video-based VLMs generally exhibit more robust temporal reasoning when asked to predict or describe future scenes, as they are explicitly trained on multi-frame or clip-level data. This gives them an inherent advantage over image-based VLMs.
>
> However, training or fine-tuning such models on large-scale video datasets requires **substantially higher computational resources** and relies on **temporal annotations** that are more difficult to obtain compared to static image data.
>
> By contrast, FutureAgent does not require any explicit video-based supervision. Through our self-supervised tuning approach, the model learns to form temporally aligned reasoning and achieves comparable robustness when asked about future scene dynamics.
>
> [7] Hwang, Jyh-Jing, et al. "Emma: End-to-end multimodal model for autonomous driving." arXiv preprint arXiv:2410.23262 (2024).
>
> [8] Jiang, Bo, et al. "Senna: Bridging large vision-language models and end-to-end autonomous driving." arXiv preprint arXiv:2410.22313 (2024).
>
> [9] Wang, Wenhai, et al. "Drivemlm: Aligning multi-modal large language models with behavioral planning states for autonomous driving." arXiv preprint arXiv:2312.09245 (2023).

---

### Official Review · Reviewer_fBK2 · 2025-10-29

**Soundness:** 3
**Presentation:** 2
**Contribution:** 2
**Rating:** 4
**Confidence:** 3

**Summary:**

This paper investigates the reliability of VLMs when applied as driving assistants, focusing on their ability to perform temporal reasoning and generate consistent responses. It identifies two critical limitations of current VLMs: response inconsistency and limited temporal reasoning. To address these issues, this paper introduce FutureVQA, a human-annotated benchmark dataset designed to evaluate future scene reasoning capabilities of VLMs, and propose a self-supervised tuning approach (FutureAgent) that enhances temporal consistency and reasoning without requiring explicit temporal labels. Experiments on multiple open-source and commercial VLMs demonstrate that the proposed method effectively improves response consistency and future scene prediction performance.

**Strengths:**

This paper analyzes and highlights key reliability issues (response inconsistency and poor temporal reasoning) of VLMs in safety-critical driving scenarios.

FutureVQA provides a valuable human-annotated dataset tailored for evaluating future scene reasoning in driving, addressing the gap in existing benchmarks that lack focus on temporal dynamics.

**Weaknesses:**

Inference speed and suitability for real-time driving applications are not discussed

The cite format is incorrect, it seems the authors used 'cite' rather than 'citep' required in the template.

FutureVQA focuses on basic future scene questions; it does not fully cover complex driving scenarios (e.g., emergency situations, multi-agent interactions), raising concerns about the benchmark’s ecological validity.

This paper identifies response inconsistency but does not deeply analyze its underlying causes (e.g., model architecture, training data biases, or prompt sensitivity mechanisms), limiting targeted improvements.

Whether there are other specialized temporal reasoning models, it is hard to assess the technical contribution and relative advantage of this paper in the driving domain.

**Questions:**

Have you explored why VLMs exhibit response inconsistency (e.g., internal randomness, prompt phrasing sensitivity, or knowledge gaps)? How can these specific causes be mitigated beyond the proposed self-supervised tuning?

How does the inference speed of FutureAgent? Can this method be applied for real-time deployment in autonomous driving systems?

Can FutureVQA include more complex driving scenarios (e.g., adverse weather, traffic accidents) and diverse question types (e.g., causal reasoning about collisions)?

---

> ### Author Response · Authors · 2025-11-22
> **Author Response (1/4)**
>
> We thank the reviewer for the very constructive and detailed feedback regarding the discussion on real-time inference, benchmark scope, analysis of inconsistency causes, and comparison to specialized temporal reasoning models. We have addressed these points below. In addition, we have corrected the citation formatting according to the template requirements.
>
> ### Question1
> > “Have you explored why VLMs exhibit response inconsistency (e.g., internal randomness, prompt phrasing sensitivity, or knowledge gaps)?”
>
> This response inconsistency is indeed a critical and challenging problem, closely tied to model explainability. To the best of our knowledge, there is still no clear conclusion regarding its root causes or effective mitigation strategies, and existing works continue to investigate this issue [2, 3, 4].
>
> Potential contributing factors include floating-point non-associativity and implementation differences across GPU kernels, as well as **sensitivity to prompt perturbations** and **random guessing due to knowledge gaps**. Although no definitive explanation has been established, these studies point to several plausible sources of variability.
>
> In our work, we focus on two primary sources of inconsistency:
>
> (1) **Prompt-perturbation sensitivity**, where the model possesses relevant knowledge and shows a clear preference across multiple trials but remains sensitive to minor input changes; and
> (2) **Random guessing**, which reflects potential knowledge gaps leading the model to respond without a consistent preference.
>
> These two forms are analyzed and formalized in **Section 3.1 (lines 144–154)**.
>
> 1. **Prompt-perturbation sensitivity**
>
>     We apply semantics-preserving perturbations $ T_{\pi}(x) $ (e.g., shuffling answer options) multiple times and prompt the model to answer the same question.
>
>     By analyzing results across multiple rounds, we observe that the model typically shows a preference toward certain answers based on its pretrained knowledge, indicating that the behavior is not uniformly random.
>
>     However, the responses remain partially inconsistent across trials, even when the temperature is reduced to zero, and in some cases, the preferred answer itself also changes.
>
>     Such failures can be identified by a non-zero total-variation distance, as defined in **Equation (2)** or by an elevated flip rate $ \text{FR}(x) $ as shown in **Equation (3)**.
>
> 2. **Random guessing**
>
>     Another source of inconsistency arises from random guessing caused by a knowledge gap. We identify this behavior by varying the number of trials and answer options. The observed performance drop scales proportionally with both the number of options and the number of trials, indicating that the model lacks a clear preference and is effectively selecting answers at random.
>
> In the table below, we report the percentage of performance drop attributable to random guessing (Rand %) and to inconsistent yet preference-biased responses (Pref %).
>
> | Model | $S-M \downarrow$ | Rand(%) | Pref (%) |
> | --- | --- | --- | --- |
> | LLV-v1.5-7b  | 21.3% | $\approx$ 19% | 2~3% |
> | Qwen-VL-7b | 20.0% | $\approx$ 18% | 2~3% |
> | Qwen2.5-VL-7b  | 10.0% | $\approx$ 8% | 2~3% |
> | GPT-4o | 11.1% | $\approx$ 8% | 3~4% |
> | FutureAgent∗ | 7.8% | $\approx$ 6% | 1~3% |
>
> ### Question 2
> > “How can these specific causes be mitigated beyond the proposed self-supervised tuning?”
>
> While our self-supervised tuning is primarily designed to strengthen temporal reasoning and align events across time, we also incorporate Chain-of-Thought (CoT) reasoning (**Section 4, Appendix B**). When reasoning about future scenes, the model is prompted to first infer the *near* future before extending its reasoning to more distant states. In **Section 5, Table 1**, the model is instructed to think about the near future before answering the question; this additional reasoning step acts as a “hint,” and empirical results show that the model produces more consistent answers. Notably, after applying our tuning and CoT, **single-trial accuracy decreases while multi-trial accuracy increases**, suggesting that the model becomes more stable and less sensitive to prompt perturbations.
>
> In conclusion, while self-supervision improves temporal alignment and strengthens temporal reasoning, we find that the primary improvement in mitigating inconsistency caused by prompt-perturbation sensitivity comes from the use of Chain-of-Thought. Our ablation study further supports this observation: without CoT, the performance gain is minimal, less than 0.2%.
>
> For the second cause, random guessing due to knowledge gaps, the most effective mitigation is to use a stronger backbone model or further fine-tune the VLM with more domain-specific knowledge. As shown in **Section 5, Table 1**, stronger base models exhibit a smaller gap between single-trial and multi-trial performance, indicating a reduced likelihood of random guessing.

---

> ### Author Response · Authors · 2025-11-22
> **Author Response (2/4)**
>
> ### Question 3
> > “How does the inference speed of FutureAgent? Can this method be applied for real-time deployment in autonomous driving systems?”
>
> We appreciate the reviewer’s thoughtful question on mitigation strategies and real-time feasibility. The real-time deployment of VLM in an real vehicle is indeed still an **open research problem**, and we would like to emphasize that reliability is equally critical in such settings.
>
> The inference speed of FutureAgent depends strongly on the underlying hardware, image resolution, and backbone model. In the table below, we report the inference performance of our approach using a single NVIDIA A100 (80 GB) GPU. Our research primarily focuses on evaluating reliability and consistency of VLM reasoning rather than real-time deployment, and therefore we did not test the system on an actual vehicle.
>
> | #image | # Tokens(Image) | # Tokens(Prompt) | Time |
> | ------ | ------ | ------ | ------ |
> | 1 | 576 | 120 | $\approx$ 0.9 s |
> | 5 | 576x5 | 120 | $\approx$ 1.7 s |
>
> However, given the observed inference speed and typical frame rates, deploying such a model on a real vehicle, which has strict time and hardware constraints, would likely require several optimizations such as:
>
> 1. **Quantization:** using 4-bit weights instead of the 16-bit precision used during training.
> 2. **Model distillation:** compressing the reasoning process into a smaller, faster model.
> 3. **Key-frame processing:** running inference only on key frames and using predictions to generate high-level driving instructions, rather than low-level, frame-by-frame control (>30 FPS). Similar arechitecture has been explored [5, 6]
>
> These optimizations could make real-time or near-real-time deployment more feasible, although achieving reliable real-time performance remains an open research challenge.
>
> ### Question 4
> > “Can FutureVQA include more complex driving scenarios (e.g., adverse weather, traffic accidents) and diverse question types (e.g., causal reasoning about collisions)?”
>
> We appreciate the reviewer’s insightful suggestion regarding the inclusion of more complex driving scenarios and diverse question types.
>
> We would like to clarify the design choice behind the benchmark. In this work, our primary goal is to evaluate the **reliability** of VLM responses in driving scenes, rather than measuring the model’s overall driving knowledge. Specifically, as illustrated in *Figure 1*, our focus is on whether the model can provide consistent and temporally aligned predictions across similar visual inputs.
>
> In the initial development stage, we experimented with a more challenging subset that included **low-light and environments that require advance scene reasoning**. However, we observed that both basic and advanced models suffered a significant and comparable performance drop under these conditions. Since the models failed to interpret the scenes **even without temporal reasoning**, further analysis of response consistency and temporal reasoning became infeasible.
> For this reason, we intentionally selected relatively simple and well-structured scenarios to isolate and analyze consistency-related factors.
>
> However, we completely agree that evaluating reliability under such challenging conditions is crucial. Combining such benchmark with our proposed benchmark, will enable a more comprehensive assessment of VLMs in driving.

---

> ### Author Response · Authors · 2025-11-22
> **Author Response (3/4)**
>
> ### Weakness 4
> > “Whether there are other specialized temporal reasoning models, it is hard to assess the technical contribution and relative advantage of this paper in the driving domain.”
>
> Comparing our model and image based VLMs to other model that specialized in temporal reasoning is indeed important. In the experiments (**Section 5 Figure 5, Table 2**), we conducted experiments using video-based VLMs (LLaVA-Video, Video-LLaMA) that are pretrained or fine-tuned with explicit temporal inputs. As expected, these models exhibit more stable temporal reasoning compared to image based VLMs due to their architecture and training that explicitly incorporate temporal cues.
>
> However, training or fine-tuning such models requires **substantial temporal annotations**, dense clip-level supervision, and significantly higher computational resources. In contrast, our proposed method does **not rely on temporal labels**, yet is able to achieve comparable performance in temporal alignment and scene-consistency evaluation. This demonstrates that FutureAgent provides an effective and annotation-efficient alternative for improving temporal reliability in VLM-based driving systems.
>
> | Model | Acc@1s | Acc@4s | Acc@12s | $ \Delta Acc^{12s}_{1s} $ | mAcc |
> | --- | --- | --- | --- | --- | --- |
> | LLaVA-Video | 53.7% | 46.5% | 43.4% | -10.3% | 46.8% |
> | Video-LLaMA | 52.4% | 41.2% | 37.2% | -15.2% | 42.4% |
> | Ours | 60.8% | 50.7% | 43.6% | -16.6% | 50.1% |
>
> [2] He, Horace and Thinking Machines Lab, "Defeating Nondeterminism in LLM Inference",
> Thinking Machines Lab: Connectionism, Sep 2025.
>
> [3] Kalai, Adam Tauman, et al. "Why language models hallucinate." arXiv preprint arXiv:2509.04664 (2025).
>
> [4] Ahn, Jihyun Janice, and Wenpeng Yin. "Prompt-reverse inconsistency: Llm self-inconsistency beyond generative randomness and prompt paraphrasing." arXiv preprint arXiv:2504.01282 (2025).
>
> [5] Jiang, Bo, et al. "Senna: Bridging large vision-language models and end-to-end autonomous driving." arXiv preprint arXiv:2410.22313 (2024).
>
> [6] Zhang, Ruifei, et al. "AdaDrive: Self-Adaptive Slow-Fast System for Language-Grounded Autonomous Driving." *Proceedings of the IEEE/CVF International Conference on Computer Vision*. 2025.

---

> > ### Comment · Reviewer_fBK2 · 2025-11-25
> >
> > Thanks for your clarification. I see the point that you can improve consistency and temporal reliability without temporal labels. However, my concern regarding the diversity and practicality of the scenes remains. Furthermore, the inference time is rather long, even when using a strong A100 GPU. These issues impair the practicality. Therefore, I regret to maintain my current score.

---

> ### Author Response · Authors · 2025-11-26
> **Author Response (4/4)**
>
> We sincerely appreciate your active engagement in the discussion and the consideration you have given to our work. We would like to clarify the two concerns you raised:
>
> 1. Regarding inference speed and practicality
>
> We fully agree that reducing inference time is critical for real-world practicality. However, achieving sub-0.1s inference, similar to conventional real-time visual detectors, is still beyond the capabilities of current VLMs. Maintaining strong reasoning performance while reaching such latency remains an **open research challenge**. Our model runs at approximately 0.9s for a single-image input and 1.7s for five frames on an A100 GPU, which is **in line with most of existing VLMs of similar scale.** Moreover, comparing with the baseline before tuning our method do not significantly increase the inference time.
>
> In this work, our primary focus is on improving temporal consistency and reliability rather than building a real-time system. We believe that our findings are equally important for advancing reliable driving assistants, where consistency and stability of predictions are crucial.
>
> | Model                | # Images | # Tokens (Prompt) | Time       |
> |---------------------|----------|-------------------|------------|
> | Ours (34b)         | 1        | 120               | ≈ 0.9 s    |
> | Ours (34b)        | 5        | 120               | ≈ 1.7 s    |
> | Baseline (34b)   | 1        | 120               | ≈ 0.9 s          |
> | Baseline (34b)  | 5        | 120               | ≈ 1.6 s          |
> | Yi-VL(34b)           | 1        | 120               | ≈   1.1s       |
>
>
> 2. Regarding diversity
>
> In our previous response, we clarified that extremely challenging questions or visually extreme scenarios do not necessarily yield more meaningful evaluation of consistency or temporal reasoning. However, this should not be interpreted as a lack of diversity in our benchmark. As detailed in **Appendix A.1**, the dataset spans **multiple cities, weather, and traffic situations**, and the QA pairs are organized into **five semantic categories** (general knowledge, relative and absolute position, hallucination detection, and traffic-rule understanding).
>
> Furthermore, **Appendix A (Tables 5 & 6)** compares our dataset with existing benchmarks and shows that ours features a **larger vocabulary range** and **more distinct QA samples**. We hope this clarifies that diversity was an intentional design choice, balanced carefully to ensure evaluative clarity.
>
> Thank you very much for the thoughtful feedback.

---

### Official Review · Reviewer_xA9B · 2025-11-01

**Soundness:** 3
**Presentation:** 3
**Contribution:** 2
**Rating:** 6
**Confidence:** 4

**Summary:**

This paper systematically evaluates the temporal reasoning and future scene prediction capabilities of VLMs in the context of autonomous driving. The authors introduce the FutureVQA benchmark, a challenging human-annotated dataset specifically designed for future scene understanding. They further propose a self-supervised fine-tuning approach that improves models’ temporal consistency and reasoning ability without requiring explicit temporal annotations. Experimental results demonstrate the limitations of existing VLMs and show that the proposed method provides significant gains in both accuracy and temporal alignment.

**Strengths:**

1. The paper addresses an important and underexplored problem of reliable temporal reasoning for VLMs in safety-critical driving scenarios.
2. The introduction of the FutureVQA benchmark fills a gap in the evaluation of future scene understanding, featuring diverse, human-annotated, and time-specific questions.
3. The proposed self-supervised fine-tuning method is practical, annotation-efficient, and yields clear improvements without requiring additional temporal data labels.

**Weaknesses:**

1. The current experiments are conducted on general-purpose VLMs and do not include domain-specific models pre-trained for autonomous driving. Since the proposed self-supervised fine-tuning method relies on the quality of pseudo-labels generated by the baseline model, it would be interesting to see whether using models with driving-specific knowledge would lead to different performance improvements.

2. The evaluations mainly focus on quantitative metrics and lack more intuitive case studies. What are the concrete improvements before and after applying the self-supervised fine-tuning approach? It would be helpful to include representative qualitative examples to support the quantitative results, which could provide clearer evidence of the method’s effectiveness.

**Questions:**

1. The current experiments are conducted on general-purpose VLMs and do not include domain-specific models pre-trained for autonomous driving. Since the proposed self-supervised fine-tuning method relies on the quality of pseudo-labels generated by the baseline model, it would be interesting to see whether using models with driving-specific knowledge would lead to different performance improvements.

2. The evaluations mainly focus on quantitative metrics and lack more intuitive case studies. What are the concrete improvements before and after applying the self-supervised fine-tuning approach? It would be helpful to include representative qualitative examples to support the quantitative results, which could provide clearer evidence of the method’s effectiveness.

---

> ### Author Response · Authors · 2025-11-22
> **Author Response (1/1)**
>
> We appreciate the reviewer’s thoughtfult feedback and recognition of the importance of reliable temporal reasoning in driving scenarios, the contribution of the FutureVQA benchmark, and the practicality of our annotation-efficient self-supervised approach. Your comments are greatly appreciated and help affirm the motivation and value of our work.
>
> ### Question1
> > “The current experiments are conducted on general-purpose VLMs and do not include domain-specific models pre-trained for autonomous driving. Since the proposed self-supervised fine-tuning method relies on the quality of pseudo-labels generated by the baseline model, it would be interesting to see whether using models with driving-specific knowledge would lead to different performance improvements.”
>
> We thank the reviewer for this thoughtful suggestion. Regarding whether richer domain-specific driving knowledge leads to different performance outcomes, our experiments show a clear distinctionfor different objective:
>
> - **Yes** — driving-specific models perform better on ***regular QA** and scene interpretation*.
>
>     Models fine-tuned on driving datasets exhibit stronger understanding of traffic semantics, which leads to higher accuracy when the visual input is directly provided.
>
> - **No** — this improvement does **not** translate to better *consistency* or *temporal reasoning*.
>
>     A key finding of our work is that a model can possess stronger driving knowledge (as reflected in improved regular QA performance; see Figure 5 and Table 1 in Section 5) yet still remain inconsistent and struggle with future-scene reasoning.
>
>
> The first evidence of this phenomenon comes from models such as GPT-4o. As shown in **Section 5 (Figure 5 and Table 2),** GPT-4o performs better than our baselines on standard QA tasks, indicating stronger driving-specific knowledge, yet it fails to maintain this advantage when reasoning about future scenes.
>
> Second, when we fine-tune our baseline model with driving-related QA[1], we observe the same pattern:
> the fine-tuned model achieves *higher regular QA accuracy* but does **not** show meaningful improvement in consistency or temporal robustness (see the table below).
>
> These results highlight that domain-specific knowledge alone does not address the core challenge we target, **improving reliability, in VLM-based future reasoning**.
>
> | Model | Regular QA | Consistaancy (S-M, S/M) | Robust future reasoning( \\( \Delta Acc^{12s}_{1s} \\), mAcc) |
> | --- | --- | --- | --- |
> | $\text{Baseline}^{\dagger}$(fine-tuned with domain specific data) | $\uparrow 2.5\%$ | Not enhanced  | Not enhanced  |
> | $\text{Baseline}^{*}$(fine-tuned with domain specific data) | $\uparrow 2.3\%$ | Not enhanced  | Not enhanced  |
>
> ### Question 2
> > “The evaluations mainly focus on quantitative metrics and lack more intuitive case studies.”
>
> We thank the reviewer for this valuable suggestion. To address this, we have added representative qualitative examples illustrating the improvements before and after self-supervised fine-tuning. These examples are included in **Appendix Section E**.
>
> [1] Sima, Chonghao, et al. "Drivelm: Driving with graph visual question answering." European conference on computer vision. Cham: Springer Nature Switzerland, 2024.

---

### Author Response · Authors · 2025-12-01

Dear Area Chair and reviewers,

We sincerely thank the Area Chair and the reviewers for their time and valuable feedback on our work, and we especially appreciate the additional effort from the Area Chair due to the recent incident on OpenReview. We understand that not all reviewers were able to participate in the discussion because of the unexpected changes in the review process during the rebuttal period. We have carefully addressed all raised concerns and believe that our responses and clarifications sufficiently resolve the questions presented.

In summary, we address the following points in the rebuttal:

1. **Reliability and performance under domain-specific settings**
   *(Response to **xA9B 1/1**)*
   We show that while domain-specific models can improve visual perception, they do not inherently enhance consistency or temporal reasoning ability.

2. **Intuitive case studies and representative qualitative examples**
   *(Response to **xA9B 1/1**)*
   We provide additional qualitative examples that highlight how our method improves temporal reasoning and stability in future scene alignment .

3. **Comparison against temporal and video-based models**
   *(Response to **fBK2 3/4, Z1qg 2/2**)*
   Results indicate that our method strengthens temporal reasoning without requiring temporal labels, which are costly and difficult to obtain.

4. **Inference time and feasibility for real-time deployment**
   *(Response to **fBK2 2/4, fBK2 4/4**)*
   While real-time deployment of powerful VLMs remains an open research challenge and is not the main focus of this work, our inference speed is comparable to models of similar scale and introduces minimal overhead relative to the baseline.

5. **Dataset diversity, scale, and temporal context coverage**
   *(Response to **fBK2 4/4, HYS6 1/1**)*
   We demonstrate that FutureVQA features a larger vocabulary, diverse question distribution, and carefully designed structure with human annotation. These characteristics support consistency and temporal reasoning evaluation beyond simple forecasting or perception.

6. **Sources of inconsistency and potential mitigation strategies**
   *(Response to **fBK2 1/4, Z1qg 1/2**)*
   We identify two primary causes of inconsistency and discuss how our approach could mitigates them.



We believe the above clarifications and analyses resolve the concerns raised, and we sincerely thank the AC for their time and consideration in evaluating our submission.

---

### Meta-Review · Area_Chair_kk1F · 2026-01-09

**Summary:**

This paper steps in with a clear and well-motivated setting: A reliable driving assistant should provide consistent responses and reasoning based on observed information. Authors introduce the FutureVQA benchmark, a challenging human-annotated dataset specifically designed for future scene understanding.

**Reviewer Concerns:**

Whilst reviewers acknowledge the importance and underexplored area of reliable reasoning for VLMs in safety-critical driving scenarios, there are major concerns raised by reviewers.

- Most cases are general-purpose VLMs and do not include domain-specific models pre-trained for autonomous driving.
- The evaluations mainly focus on quantitative metrics and lack more intuitive case studies.
- Lack of experiments, e.g. Inference speed and suitability for real-time driving applications.
- Limited technical settings. This reviewer further responded with "diversity and practicality of the scenes remains.", and inference latency.
- Problem formulation. The future is conditional on the ego-vehicle's own actions.
- The scale and context limitations.
- Evaluation may be judge-biased.

**Reviewer Scores:**

Authors have done a comprehensive work to address most of the concerns, adding more experiments and clarity regarding the technical details. AC read the paper, review, rebuttal. In particular, the limitted scale and context limitations still remains, although authors address them in the rebuttal. The problem setup seems not 100% relevent to the autonomous driving field, which is raised by most reviewers. Authors are encouraged to re-formulate the settings and add more detailed discussions on the topic.

---

### Decision · Program_Chairs · 2026-01-26

Reject